# Signal denoising through topographic modularity of neural circuits

**Barna Zajzon[1,2]\*, David Dahmen[1], Abigail Morrison[1,3], Renato Duarte[1,4]**

[1]Institute of Neuroscience and Medicine (INM-6) and Institute for Advanced Simulation (IAS-6) and JARA-BRAIN Institute I, Jülich Research Centre, Jülich, Germany; [2]Department of Psychiatry, Psychotherapy and Psychosomatics, RWTH Aachen University, Aachen, Germany; [3]Department of Computer Science 3 - Software Engineering, RWTH Aachen University, Aachen, Germany; [4]Donders Institute for Brain, Cognition and Behavior, Radboud University Nijmegen, Nijmegen, Netherlands

**Abstract** Information from the sensory periphery is conveyed to the cortex via structured projection pathways that spatially segregate stimulus features, providing a robust and efficient encoding strategy. Beyond sensory encoding, this prominent anatomical feature extends throughout the neocortex. However, the extent to which it influences cortical processing is unclear. In this study, we combine cortical circuit modeling with network theory to demonstrate that the sharpness of topographic projections acts as a bifurcation parameter, controlling the macroscopic dynamics and representational precision across a modular network. By shifting the balance of excitation and inhibition, topographic modularity gradually increases task performance and improves the signal-to-noise ratio across the system. We demonstrate that in biologically constrained networks, such a denoising behavior is contingent on recurrent inhibition. We show that this is a robust and generic structural feature that enables a broad range of behaviorally relevant operating regimes, and provide an in-depth theoretical analysis unraveling the dynamical principles underlying the mechanism.

**\*For correspondence:** b.zajzon@fz-juelich.de

**Competing interest:** The authors declare that no competing interests exist.

## Editor's evaluation

This manuscript puts forward a new idea that topography in neural networks helps to remove noise from inputs. The authors show that there is a critical level of topography that is needed for network to denoise inputs.

## Introduction

Sensory inputs are often ambiguous, noisy, and imprecise. Due to volatility in the environment and inaccurate peripheral representations, the sensory signals that arrive at the neocortical circuitry are often incomplete or corrupt (*Faisal et al., 2008*; *Renart and Machens, 2014*). However, from these noisy input streams, the system is able to acquire reliable internal representations and extract relevant computable features at various degrees of abstraction (*Friston, 2005*; *Okada et al., 2010*; *DiCarlo et al., 2012*). Sensory perception in the mammalian neocortex thus relies on efficiently detecting the relevant input signals while minimizing the impact of noise.

Making sense of the environment also requires the estimation of features not explicitly represented by low-level sensory inputs. These inferential processes (*Młynarski and Hermundstad, 2018*; *Parr et al., 2019*) rely on the propagation of internal signals such as expectations and predictions, the accuracy of which must be evaluated against the ground truth, that is the sensory input stream. In a highly dynamic environment, this translates to a continuous process whose precision hinges on the fidelity with which external stimuli are encoded in the neural substrate. Additionally, as the system is

modular and hierarchical (strikingly so in the sensory and motor components; *Meunier et al., 2010*; *Park and Friston, 2013*), it is critical that the external signal permeates the different processing modules despite the increasing distance from the sensory periphery (the input source) and the various transformations it is exposed to along the way, which degrade the signal via the interference of task-irrelevant and intrinsic, ongoing activity.

Accurate signal propagation can be achieved in a number of ways. One obvious solution is the direct routing and distribution of the signal, such that direct sensory input can be fed to different processing modules, which may be partially achieved through thalamocortical projections (*Sherman and Guillery, 2002*; *Nakajima and Halassa, 2017*). Another possibility, which we explore in this study, is to propagate the input signal through tailored pathways that route the information throughout the system, allowing different processing stages to retrieve it without incurring much representational loss. Throughout the mammalian neocortex, the existence and characteristics of structured projections (topographic maps) present a possible substrate for such signal routing. By preserving the relative organization of tuned neuronal populations, such maps imprint spatiotemporal features of (noisy) sensory inputs onto the cortex (*Kaas, 1997*; *Bednar and Wilson, 2016*; *Wandell and Winawer, 2011*). In a previous study (*Zajzon et al., 2019*), we discovered that structured projections can create feature-specific pathways that allow the external inputs to be faithfully represented and propagated throughout the system, but it remains unclear which connectivity properties are critical and what the underlying mechanism is. Moreover, beyond mere sensory representation, there is evidence that such structure-preserving mappings are also involved in more complex cognitive processes in associative and frontal areas (*Hagler and Sereno, 2006*; *Silver and Kastner, 2009*; *Patel et al., 2014*), suggesting that topographic maps are a prominent structural feature of cortical organization.

In this study, we hypothesize that structured projection pathways allow sensory stimuli to be accurately reconstructed as they permeate multiple processing modules. We demonstrate that, by modulating effective connectivity and regional E/I balance, topographic projections additionally serve a *denoising* function, not merely allowing the faithful propagation of input signals, but systematically improving the system's internal representations and increasing signal-to-noise ratio. We identify a critical threshold in the degree of modularity in topographic projections, beyond which the system behaves effectively as a denoising autoencoder (note that the parallel is established here on conceptual, not formal, grounds as the system is capable of retrieving the original, uncorrupted input from a noisy source, but bears no formal similarity to denoising autoencoder algorithms). Additionally, we demonstrate that this phenomenon is robust, with the qualitative behavior persisting across very different models. Theoretical considerations and network simulations show that it hinges solely on the modularity of topographic projections and the presence of recurrent inhibition, with the external input and single-neuron properties influencing where/when, but not if, denoising occurs. Our results suggest that modular structure in feedforward projection pathways can have a significant effect on the system's qualitative behavior, enabling a wide range of behaviorally relevant and empirically supported dynamic regimes. This allows the system to: (1) maintain stable representations of multiple stimulus features (*Andersen et al., 2008*); (2) amplify features of interest while suppressing others through winner-takes-all (WTA) mechanisms (*Douglas and Martin, 2004*; *Carandini and Heeger, 2011*); and (3) dynamically represent different stimulus features as stable and metastable states and stochastically switch among active representations through a winnerless competition (WLC) effect (*McCormick, 2005*; *Rabinovich et al., 2008*; *Rost et al., 2018*).

Our key finding, that the modulation of information processing dynamics and the fidelity of stimulus/feature representations results from the structure of topographic feedforward projections, provides new meaning and functional relevance to the pervasiveness of these projection maps throughout the mammalian neocortex. Beyond routing feature-specific information from sensory transducers through brainstem, thalamus, and into primary sensory cortices (notably tonotopic, retinotopic, and somatotopic maps), their maintenance within the neocortex (*Patel et al., 2014*) ensures that even cortical regions that are not directly engaged with the sensory input (higher-order cortex), can receive faithful representations of it, and that these internal signals, emanating from lower-order cortical areas, can dramatically skew and modulate the circuit's E/I balance and local functional connectivity, resulting in fundamental differences in the systems' responsiveness.

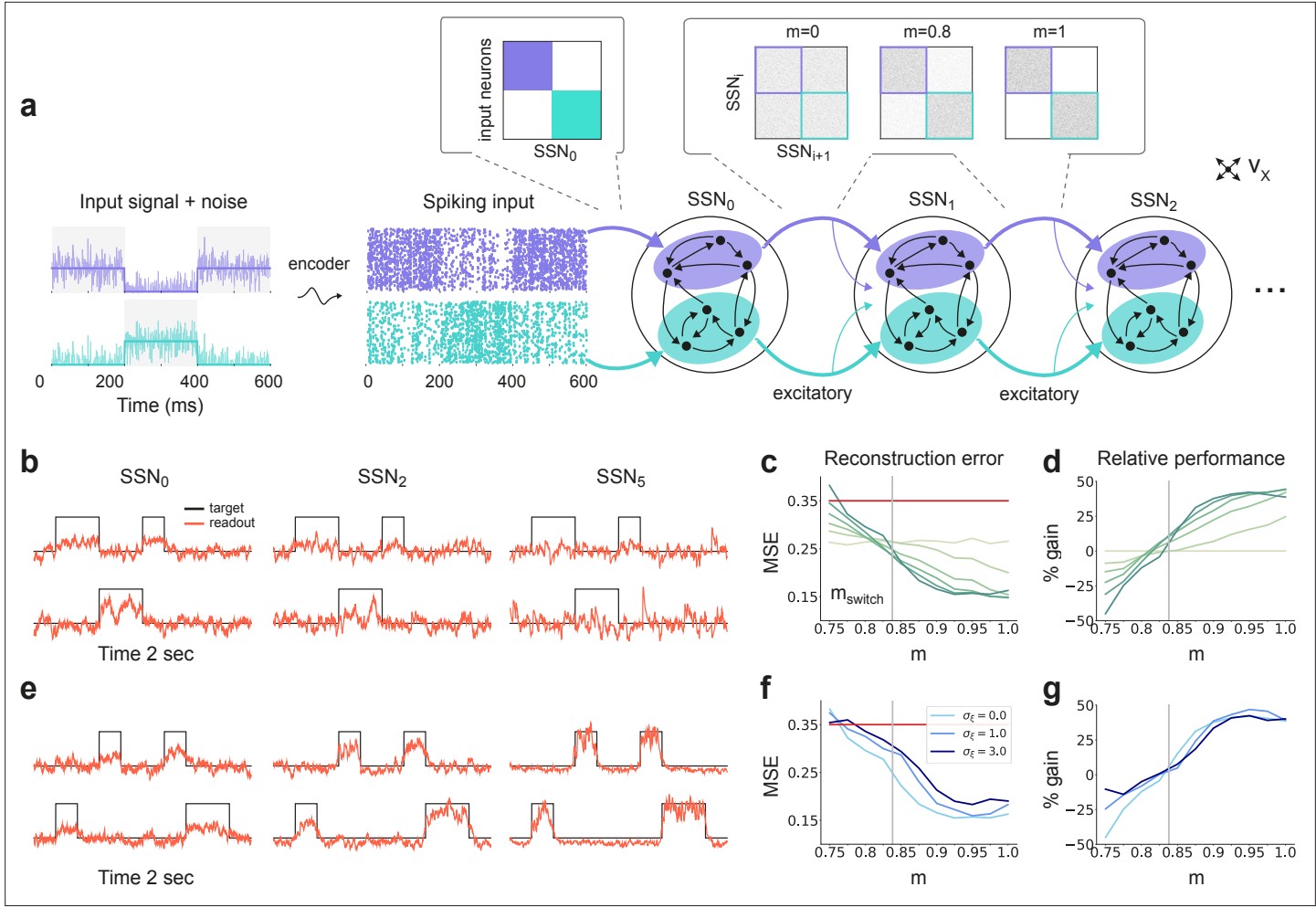

**Figure 1.** Sequential denoising spiking architecture. (**a**) A continuous step signal is used to drive the network. The input is spatially encoded in the first sub-network (SSN$_0$), whereby each input channel is mapped exclusively onto a sub-population of stimulus-specific excitatory and inhibitory neurons (schematically illustrated by the colors; see also inset, top left). This exclusive encoding is retained to variable degrees across the network, through topographically structured feedforward projections (inset, top right) controlled by the modularity parameter $m$ (see Materials and methods). This is illustrated explicitly for both topographic maps (purple and cyan arrows). Projections between SSNs are purely excitatory and target both excitatory and inhibitory neurons. (**b**) Signal reconstruction across the network. Single-trial illustration of target signal (black step function) and readout output (red curves) in three different SSNs, for $m = 0.75$ and no added noise ($\sigma_\xi = 0$). For simplicity, only two out of ten input channels are shown. (**c**) Signal reconstruction error in the different SSNs for the no-noise scenario shown in (**b**). Color shade denotes network depth, from SSN$_0$ (lightest) to SSN$_5$ (darkest). The horizontal red line represents chance level, while the gray vertical line marks the transition (switching) point $m_{\text{switch}} \approx 0.83$ (see main text). *Figure 1—figure supplement 1* shows the task performance for a broader range of parameters. (**d**) Performance gain across the network, relative to SSN$_0$, for the setup illustrated in (**b**). (**e**) as in (**b**) but for $m = 0.9$. (**f**) Reconstruction error in SSN$_5$ for the different noise intensities. Horizontal and vertical dashed lines as in (**c**). (**g**) Performance gain in SSN$_5$, relative to SSN$_0$.

The online version of this article includes the following source data and figure supplement(s) for figure 1:

**Source data 1.** Code and data for *Figure 1* and related figure supplements.

**Figure supplement 1.** Sequential denoising effect.

## Results

To investigate the role of structured pathways between processing modules in modulating the fidelity of stimulus representations, we study a network comprising up to six sequentially connected sub-networks (SSNs, see Materials and methods and *Figure 1a*). Each SSN is a *balanced random network* (see e.g. *Brunel, 2000*) of 10,000, sparsely and randomly coupled leaky integrate-and-fire (LIF) neurons (80% excitatory and 20% inhibitory). In each SSN, neurons are assigned to sub-populations associated with a particular stimulus. Excitatory neurons belonging to such stimulus-specific sub-populations then

project to the subsequent SSN with a varying degree of specificity. We refer to a set of stimulus-specific sub-populations across the network and the structured feedforward projections among them as a *topographic map*. The specificity of the map is determined by the degree of *modularity* of the corresponding projections matrices (see e.g. *Figure 1a*). Modularity is thus defined as the relative density of connections within a stimulus-specific pathway (i.e., connecting sub-populations associated to the *same* stimulus; see Materials and methods and *Figure 1a*). In the following, we study the role of topographic specificity in modulating the system's functional and representational dynamics and its ability to cope with noise-corrupted input signals.

## Sequential denoising through structured projections

By systematically varying the degree of modular specialization in the feedforward projections (modularity parameter, $m$, see Materials and methods and *Figure 1*), we can control the segregation of stimulus-specific pathways across the network and investigate how it influences the characteristics of neural representations as the signal propagates. If the feedforward projections are unstructured or moderately structured ($m \lesssim 0.8$), information about the input fails to permeate the network, resulting in a chance-level reconstruction accuracy in the last sub-network, $SSN_5$, even in the absence of noise (see *Figure 1b, c*). However, as $m$ approaches a switching value $m_{switch} \approx 0.83$, there is a qualitative transition in the system's behavior, leading to a consistently higher reconstruction accuracy across the sub-networks (*Figure 1b–e*), regardless of the amount of noise added to the signal (*Figure 1f, g*).

Beyond this transition point, reconstruction accuracy improves with depth, that is the signal is more accurately represented in $SSN_5$ than in the initial sub-network, $SSN_0$, with an effective accuracy gain of over 40% (*Figure 1d, g*). While the addition of noise does impair the absolute reconstruction accuracy in all cases (see *Figure 1—figure supplement 1*), the denoising effect persists even if the input is severely corrupted ($\sigma_\xi = 3$, see *Figure 1f, g*). This is a counter-intuitive result, suggesting that topographic modularity is not only necessary for reliable communication across multiple populations (see *Zajzon et al., 2019*), but also supports an effective denoising effect, whereby representational precision increases with depth, even if the signal is profoundly distorted by noise.

## Noise suppression and response amplification

The sequential denoising effect observed beyond the transition point $m_{switch} \approx 0.83$ results in an increasingly accurate input encoding through progressively more precise internal representations. In general, such a phenomenon could be achieved either through noise suppression, stimulus-specific response amplification or both. In this section, we examine these possibilities by analyzing and comparing the input-driven dynamics of the different sub-networks. The strict segregation of stimulus-specific sub-populations in $SSN_0$ is only fully preserved across the system if $m = 1$, in which case signal encoding and transmission primarily rely on this spatial segregation. Spiking activity across the different SSNs (*Figure 2a*) demonstrates that the system gradually sharpens the segregation of stimulus-specific sub-populations; indeed, in systems with fully modular feedforward projections, activity in the last sub-network is concentrated predominantly in the stimulated sub-populations. This effect can be observed in both excitatory (E) and inhibitory (I) populations, as both are equally targeted by the feedforward excitatory projections. The sharpening effect consists of both *noise suppression* and *response amplification* (*Figure 2b*), measured as the relative firing rates of the non-stimulated $\nu_5^{NS}/\nu_0^{NS}$ and stimulated sub-populations $\nu_5^S/\nu_0^S$, respectively. For ,$m < m_{switch}$. noise suppression is only marginal and responses within the stimulated pathways are not amplified ($\nu_5^S/\nu_0^S < 1$).

Mean-field analysis of the stationary network activity (see Materials and methods and Appendix B) predicts that the firing rates of the stimulus-specific sub-populations increase systematically with modularity, whereas the untuned neurons are gradually silenced (*Figure 2c*, left). At the transition point $m_{switch} \approx 0.83$, mean firing rates across the different sub-networks converge, which translates into a globally uniform signal encoding capacity, corresponding to the zero-gain convergence point in *Figure 1d, g*. As the degree of modularity increases beyond this point, the self-consistent state is lost again as the functional dynamics across the network shifts toward a gradual response sharpening, whereby the activity of stimulus-tuned neurons become increasingly dominant (*Figure 2a–c*). The effect is more pronounced for the deeper sub-networks. Note that the analytical results match well with those obtained by numerical simulation (*Figure 2c*, right).

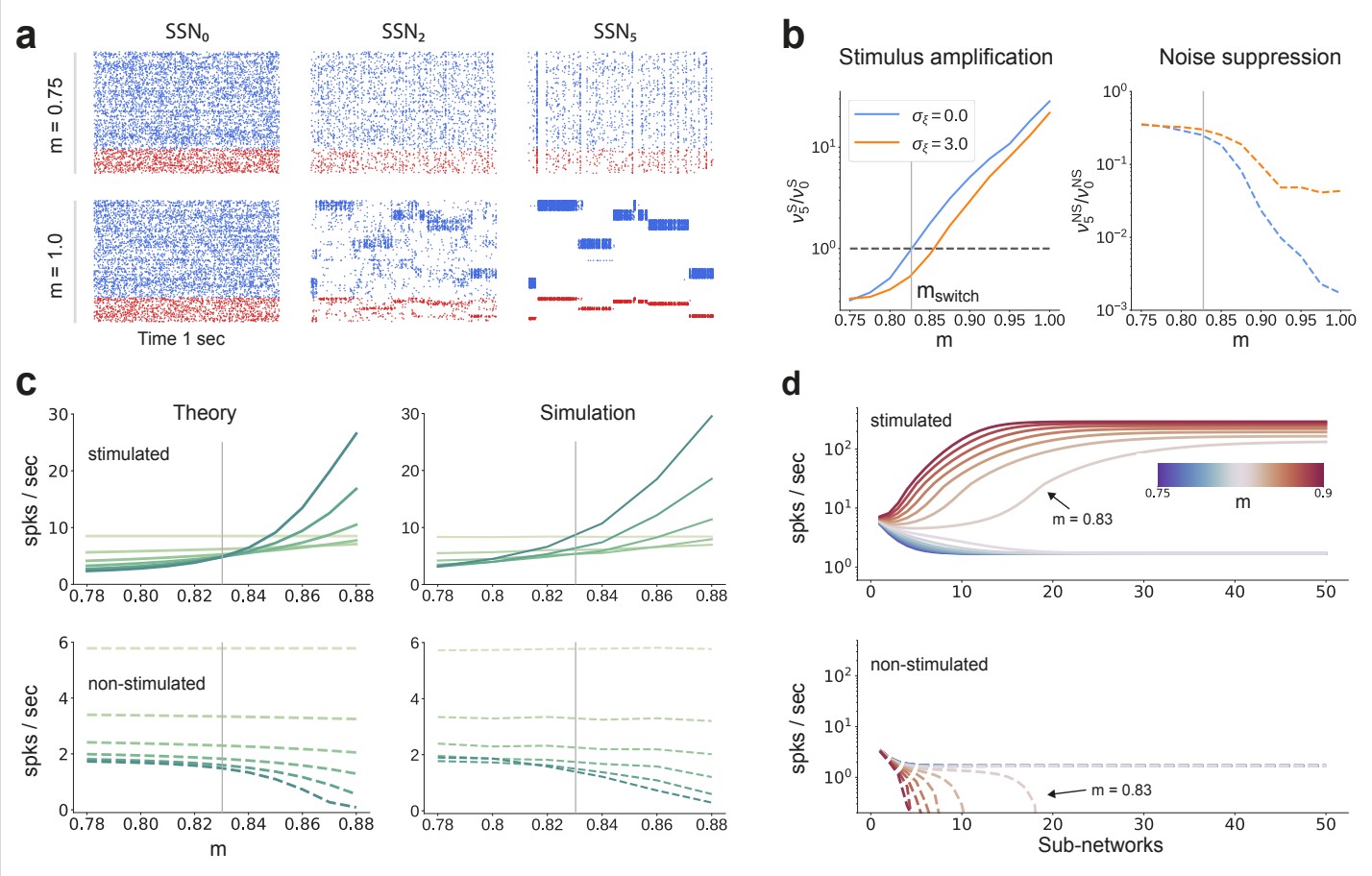

**Figure 2.** Activity modulation and representational precision. (**a**) One second of spiking activity observed across 1000 randomly chosen excitatory (blue) and inhibitory (red) neurons in $SSN_0$, $SSN_2$ and $SSN_5$, for $\sigma_\xi = 3$ and $m = 0.75$ (top) and $m = 1$ (bottom). (**b**) Mean quotient of firing rates in $SSN_5$ and $SSN_0$ ($\nu_5/\nu_0$) for stimulated (S, left) and non-stimulated (NS, right) sub-populations for different input noise levels, describing response amplification and noise suppression, respectively. (**c**) Mean firing rates of the stimulated (top) and non-stimulated (bottom) excitatory sub-populations in the different SSNs (color shade as in *Figure 1*), for $\sigma_\xi = 0$. For modularity values facilitating an asynchronous irregular regime across the network, the firing rates predicted by mean-field theory (left) closely match the simulation data (right). (**d**) Mean-field predictions for the stationary firing rates of the stimulated (top) and non-stimulated (bottom) sub-populations, in a system with 50 sub-networks and $\sigma_\xi = 0$. Note that all reported simulation data correspond to the mean firing rates acquired over a period of 10 s and averaged across 5 trials per condition. *Figure 2—figure supplement 1* shows the firing rates as a function of the input intensity $\lambda$.

The online version of this article includes the following source data and figure supplement(s) for figure 2:

**Source data 1.** Code and data for *Figure 2* and related figure supplements.

**Figure supplement 1.** Mean-field predictions for the gain in the firing rates of stimulated sub-populations.

In the limit of very deep networks (up to 50 SSNs, *Figure 2d*) the system becomes bistable, with rates converging to either a high-activity state associated with signal amplification or a low-activity state driven by the background input. The transition point is observed at a modularity value of $m = 0.83$, matching the results reported so far. Below this value, elevated activity in the stimulated sub-populations can be maintained across the initial sub-networks (<10), but eventually dies out; the rate of all neurons decays and information about the input cannot reach the deeper populations. Importantly, for $m = 0.83$, the transition toward the high-activity state is slower. This allows the input signal to faithfully propagate across a large number of sub-networks ($\approx 15$), without being driven into implausible activity states.

## E/I balance and asymmetric effective couplings

The departure from the balanced activity in the initial sub-networks can be better understood by zooming in at the synaptic level and analyzing how topography influences the synaptic input currents.

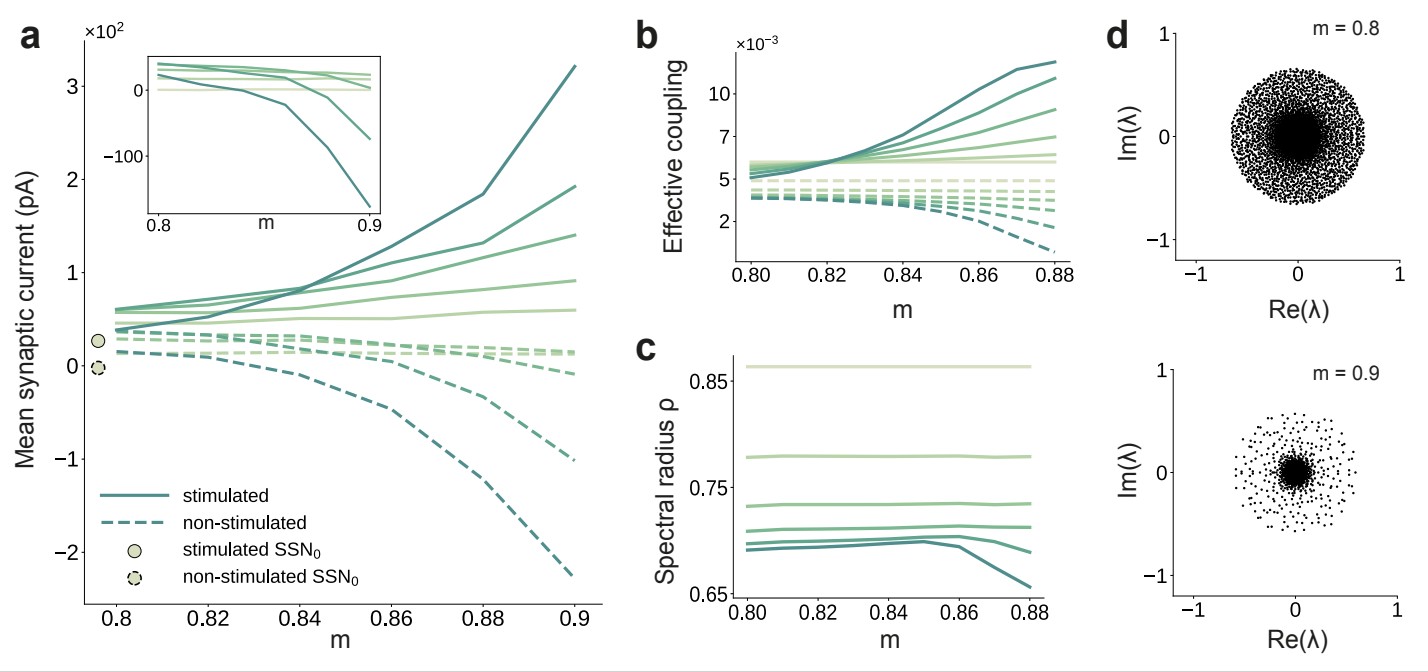

**Figure 3.** Asymmetric effective couplings modulate the E/I balance and support sequential denoising. (**a**) Mean synaptic input currents for neurons in the stimulated (solid curves) and non-stimulated (dashed curves) excitatory sub-populations in the different SSNs. To avoid clutter, data for $SSN_0$ are only shown by markers (independent of $m$). Inset shows the currents (in pA) averaged over all excitatory neurons in the different sub-networks; increasing modularity leads to a dominance of inhibition in the deeper sub-networks. Color shade represents depth, from $SSN_1$ (light) to $SSN_5$ (dark). (**b**) Mean-field approximation of the effective recurrent weights in $SSN_5$. Curve shade and style as in (**a**). (**c**) Spectral radius of the effective connectivity matrices $\rho(W)$ as a function of modularity. (**d**) Eigenvalue spectra for the effective coupling matrices in $SSN_5$, for $m = 0.8$ (top) and $m = 0.9$ (bottom). The largest negative eigenvalue (outlier, see Materials and methods), characteristic of inhibition-dominated networks, is omitted for clarity.

The online version of this article includes the following source data for figure 3:

**Source data 1.** Code and data for *Figure 3*.

The segregation of feedforward projections into stimulus-specific pathways breaks the symmetry between excitation and inhibition (see *Figure 3a*) that characterizes the balanced state (*Haider et al., 2006*; *Shadlen and Newsome, 1994*), for which the first two sub-networks were tuned (see Materials and methods). E/I balance is thus systematically shifted toward excitation in the stimulated populations and inhibition in the non-stimulated ones. Neurons belonging to sub-populations associated with the active stimulus receive significantly more net overall excitation, whereas the other neurons become gradually more inhibited. This disparity grows not only with modularity but also with network depth. Overall, across the whole system, increasing modularity results in an increasingly inhibition-dominated dynamical regime (inset in *Figure 3a*), whereby stronger effective inhibition silences non-stimulated populations, thus sharpening stimulus/feature representations by concentrating activity in the stimulus-driven sub-populations.

To gain an intuitive understanding of these effects from a dynamical systems perspective, we linearize the network dynamics around the stationary working points of the individual populations (*Tetzlaff et al., 2012*) in order to obtain the effective connectivity $W$ of the system (see Materials and methods and Appendix B). The effective impact of a single spike from a presynaptic neuron $j$ on the firing rate of a postsynaptic neuron $i$ (the effective weight $w_{ij} \in W$) is determined not only by the synaptic efficacies $J_{ij}$, but also by the statistics of the synaptic input fluctuations to the target cell $i$ that determine its excitability (see Materials and methods, *Equation 6*). This analysis reveals that there is an increase in the effective synaptic input onto neurons in the stimulated sub-populations as a function of modularity (*Figure 3b*). Conversely, non-stimulated neurons effectively receive weaker excitatory (and stronger inhibitory) drive and become increasingly less responsive (see *Figure 3a, b*). The role of topographic modularity in denoising can thus be understood as a transient, stimulus-specific change in effective connectivity.

For low and moderate topographic precision ($m \lesssim 0.83$), denoising does not occur as the effective weights are sufficiently similar to maintain a stable E/I balance across all populations and sub-networks (*Figure 3a, b*), resulting in a relatively uniform global dynamical state (indicated in *Figure 3c* by a constant spectral radius for $m \lesssim 0.83$, see also Materials and methods) and stable linearized dynamics ($\rho(W) < 1$).

However, as the feedforward projections become more structured, the system undergoes qualitative changes: after a weak transient ($0.83 \lesssim m \lesssim 0.85$) the spectral radius $\rho$ in the deep SSNs expands due to the increased effective coupling to the stimulated sub-population (*Figure 3b*); the spectral radius eventually ($m \gtrsim 0.85$) contracts with increasing modularity (*Figure 3c, d*). Given that $\rho$ is determined by the variance of $W$, that is heterogeneity across connections (*Rajan and Abbott, 2006*), this behavior is expected: most weights are in the non-stimulated pathways, which decrease with larger $m$ and network depth (*Figure 3b*). Strong inhibitory currents (*Figure 3a*) suppress the majority of neurons, thereby reducing noise, as demonstrated by the collapse of the bulk of the eigenvalues toward the center for larger $m$ (*Figure 3d*). Indicative of a more constrained state space, this contractive effect suggests that population activity becomes gradually entrained by the spatially encoded input along the stimulated pathway, whereas the responses of the non-stimulated neurons have a diminishing influence on the overall behavior.

By biasing the effective connectivity of the system, precise topography can thus modulate the balance of excitation and inhibition in the different sub-networks, concentrating the activity along specific pathways. This results in both a systematic amplification of stimulus-specific responses and a systematic suppression of noise (*Figure 2b*). The sharpness/precision of topographic specificity along these pathways thus acts as a critical control parameter that largely determines the qualitative behavior of the system and can dramatically alter its responsiveness to external inputs.

## Modulating inhibition

How can the system generate and maintain the elevated inhibition underlying such a noise-suppressing regime? On the one hand, feedforward excitatory input may increase the activity of certain excitatory neurons in $E_i$ of sub-network $SSN_i$, which, in turn, can lead to increased mean inhibition through local recurrent connections. On the other hand, denoising could depend strongly on the concerted topographic projections onto $I_i$. Such structured feedforward inhibition is known to play important

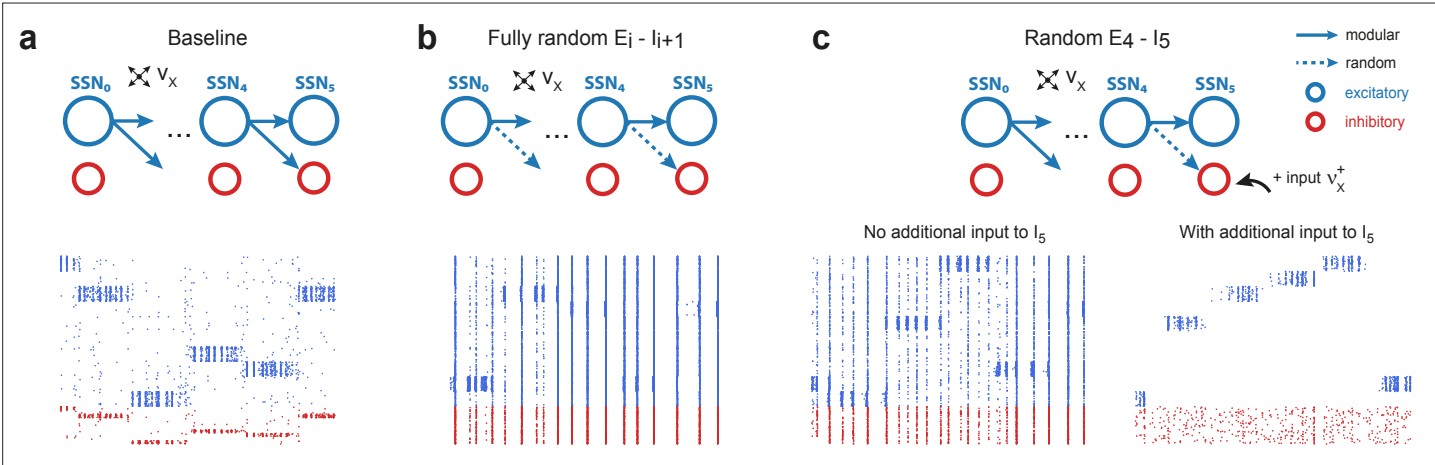

**Figure 4.** Modular projections to inhibitory populations stabilize network dynamics. Raster plots show 1 s of spiking activity of 1000 randomly chosen neurons in $SSN_5$, for different network configurations. (**a**) Baseline network with $m = 0.88$. (**b**) Unstructured feedforward projections to the inhibitory sub-populations lead to highly synchronized network activity, hindering signal representation. (**c**) Same as the baseline network in (**a**), but with random projections for $E_4 \rightarrow I_5$ and additional but unspecific (Poissonian) excitatory input to $I_5$ controlled via $\nu_X^+$. Without such input ($\nu_X^+ = 0$, left), the activity is strongly synchronous, but this is compensated for by the additional excitation, reducing synchrony and restoring the denoising property ($\nu_X^+ = 10$ spikes/s, right). *Figure 4—figure supplement 1* depicts the activity statistics in the last two modules, for the different scenarios.

The online version of this article includes the following source data and figure supplement(s) for figure 4:

**Source data 1.** Code and data for *Figure 4* and related figure supplements.

**Figure supplement 1.** Spiking statistics for different feedforward wiring to inhibitory neurons.

functional roles in, for example, sharpening the spatial contrast of somatosensory stimuli (*Mountcastle and Powell, 1959*) or enhancing coding precision throughout the ascending auditory pathways (*Roberts et al., 2013*).

To investigate whether recurrent activity alone can generate sufficiently strong inhibition for signal transmission and denoising, we maintained the modular structure between the excitatory populations and randomized the feedforward projections onto the inhibitory ones ($m = 0$ for $E_i \rightarrow I_{i+1}$, compare top panels of *Figure 4a, b*). This leads to unstable firing patterns in the downstream sub-networks, characterized by significant accumulation of synchrony and increased firing rates (see bottom panels of *Figure 4a, b* and *Figure 4—figure supplement 1a, b*). These effects, known to result from shared pre-synaptic excitatory inputs (see e.g. *Shadlen and Newsome, 1998*; *Tetzlaff et al., 2003*; *Kumar et al., 2008a*), are more pronounced for larger $m$ and network depth (see *Figure 4—figure supplement 1*). Compared with the baseline network, whose activity shows clear spatially encoded stimuli (sequential activation of stimulus-specific sub-populations [*Figure 4a*, bottom]), removing structure from the projections onto inhibitory neurons abolishes the effect and prevents accurate signal transmission.

These effects of unstructured inhibitory projections are so marked that they can be observed even if a single set of projections is modified: this can be seen in *Figure 4c*, where only the $E_4 \rightarrow I_5$ connections are randomized. It is worth noting, however, that the excessive synchronization that results from unstructured inhibitory projections (*Figure 4c*, bottom left, no additional input condition) can be easily counteracted by driving $I_5$ (the inhibitory population that receives only unstructured projections) with additional uncorrelated external input. If strong enough ($\nu_X^+ \approx 10\text{spk/sec}$), this additional external drive pushes the inhibitory population into an asynchronous regime that restores the sharp, stimulus-specific responses in the excitatory population of the corresponding sub-network (see *Figure 4c*, bottom right, and *Figure 4—figure supplement 1c*).

These results emphasize the control of inhibitory neurons' responsiveness as the main causal mechanism behind the effects reported. Elevated local inhibition is strictly required, but whether this is achieved by tailored, stimulus-specific activation of inhibitory sub-populations, or by uncorrelated excitatory drive onto all inhibitory neurons appears to be irrelevant and both conditions result in sharp, stimulus-tuned responses in the excitatory populations.

## A generalizable structural effect

We have demonstrated that, by controlling the different sub-networks' operating point, the sharpness of feedforward projections allows the architecture to systematically improve the quality of internal representations and retrieve the input structure, even if profoundly corrupted by noise. In this section, we investigate the robustness of the phenomenon in order to determine whether it can be entirely ascribed to the topographic projections (a structural/architectural feature) or if the particular choices of models and model parameters for neuronal and synaptic dynamics contribute to the effect.

To do so, we study two alternative model systems on the signal denoising task. These are structured similar to the baseline system explored so far, comprising separate sequential sub-networks with modular feedforward projections among them (see *Figure 1* and Materials and methods), but vary in total size, neuronal and synaptic dynamics. In the first test case, only the models of synaptic transmission and corresponding parameters are altered. To increase biological verisimilitude and following *Zajzon et al., 2019*, synaptic transmission is modeled as a conductance-based process, with different kinetics for excitatory and inhibitory transmission, corresponding to the responses of $\text{AMPA}$ and $\text{GABA}_a$ receptors, respectively, see Materials and methods and *Supplementary file 3* for details. The results, illustrated in *Figure 5a*, demonstrate that task performance and population activity across the network follow a similar trend to the baseline model (*Figures 1 and 2a, b*). Despite severe noise corruption, the system is able to generate a clear, discernible representation of the input as early as $\text{SSN}_2$ and can accurately reconstruct the signal. Importantly, the relative improvement with increasing modularity and network depth is retained. In comparison to the baseline model, the transition occurs for a slightly different topographic configuration, $m_{\text{switch}} \approx 0.85$, at which point the network dynamics converges toward a low-rate, stable asynchronous irregular regime across all populations, facilitating a linear firing rate propagation along the topographic maps (*Figure 5—figure supplement 1*).

The second test case is a smaller and simpler network of nonlinear rate neuron models (see *Figure 5b* and Materials and methods) which interact via continuous signals (rates) rather than

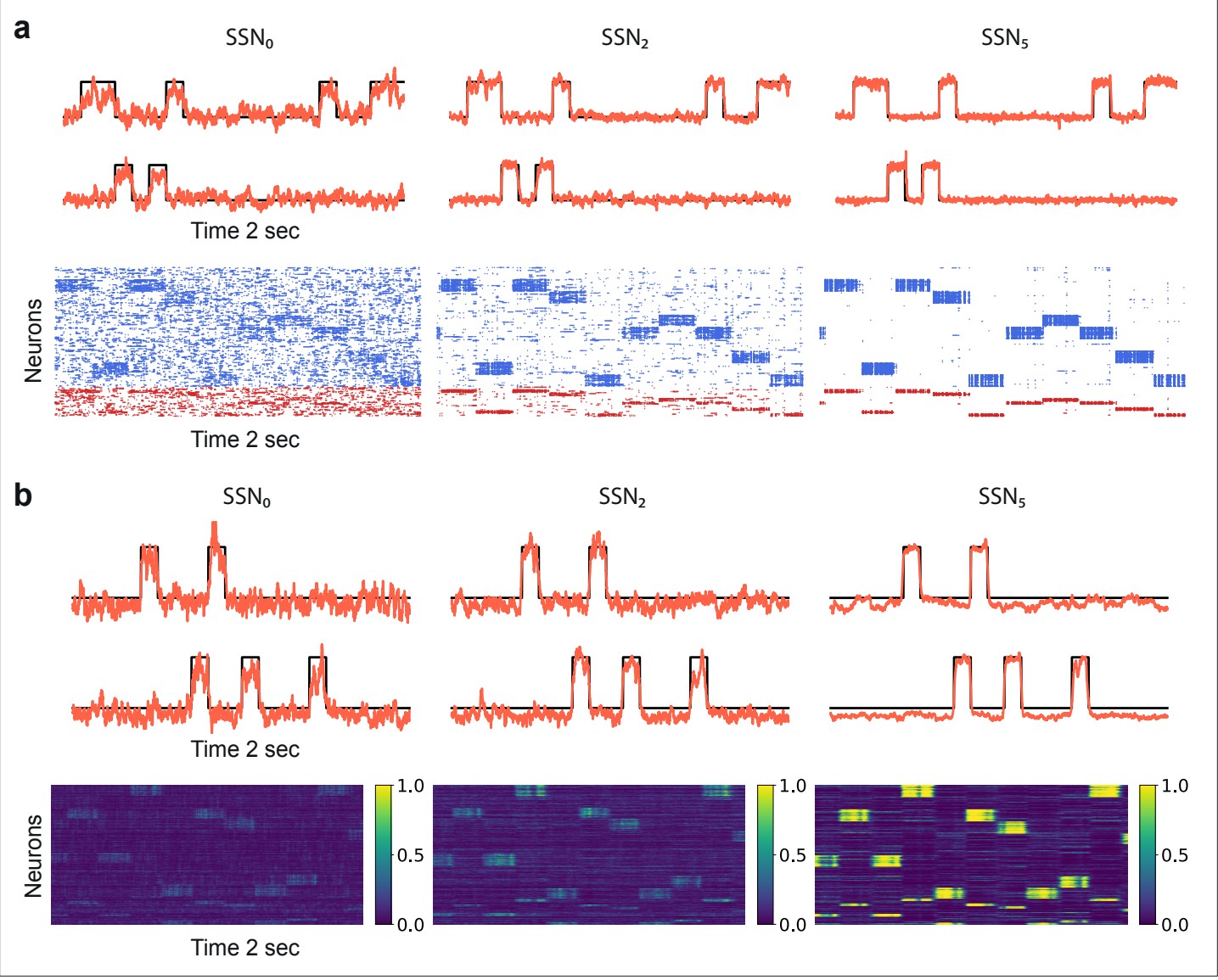

**Figure 5.** Denoising through modular topography is a robust structural effect. (**a**) Signal reconstruction (top) and corresponding network activity (bottom) for a network with leaky integrate-and-fire (LIF) neurons and conductance-based synapses (see Materials and methods). Single-trial illustration of target signal (black step function) and readout output (red curves) in three different SSNs, for $m = 0.9$ and strong noise corruption ($\sigma_\xi = 3$). For simplicity, only two out of ten input channels are shown. *Figure 5—figure supplement 1* shows additional activity statistics. (**b**) As in (**a**) for a rate-based model with $m = 1$ and $\sigma_\xi = 1$ (see Materials and methods for details).

The online version of this article includes the following source data and figure supplement(s) for figure 5:

**Source data 1.** Code and data for *Figure 5* and related figure supplements.

**Figure supplement 1.** Spiking statistics for the conductance-based model.

discontinuities (spikes). Despite these profound differences in the neuronal and synaptic dynamics, the same behavior is observed, demonstrating that sequential denoising is a structural effect, dependent on the population firing rates and thus less sensitive to fluctuations in the precise spike times. Moreover, the robustness with respect to the network size suggests that denoising could also be performed in smaller, localized circuits, possibly operating in parallel on different features of the input stimuli.

## Variable map sizes

Despite their ubiquity throughout the neocortex, the characteristics of structured projection pathways is far from uniform (*Bednar and Wilson, 2016*), exhibiting marked differences in spatial precision

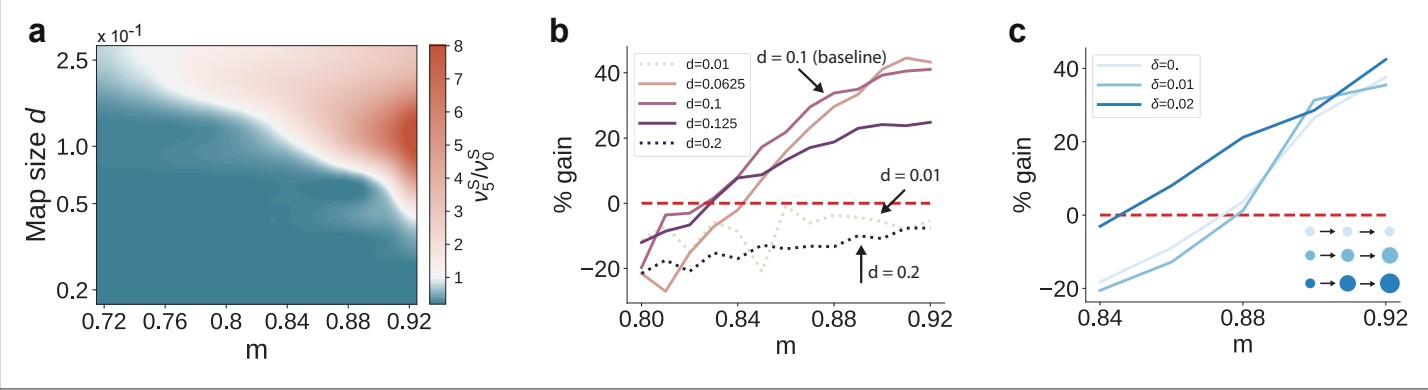

**Figure 6.** Variation in the map sizes. (**a**) Ratio of the firing rates of the stimulated sub-populations in the first and last sub-networks, $\nu_5^S/\nu_0^S$, as a function of modularity and map size (parameterized by $d$ and constant throughout the network, that is $\delta = 0$, see Materials and methods). Depicted values correspond to stationary firing rates predicted by mean-field theory, smoothed using a Lanczos filter. Note that, in order to ensure that every neuron was uniquely tuned, that is there is no overlap between stimulus-specific sub-populations, the number of sub-populations was igen chosen to be proportional to the map size ($N_C = 1/d$). (**b, c**) Performance gain in SSN$_5$ relative to SSN$_0$ (ten stimuli, as in **Figure 1d, g**), for varying properties of structural mappings: (**b**) fixed map size ($\delta = 0$) with color shade denoting map size, and (**c**) linearly increasing map size ($\delta > 0$) and a smaller initial map size $d_0 = 0.04$. The results depict the average performance gains measured across five trials, using the current-based model illustrated in **Figure 1** (ten stimuli) and no input noise ($\sigma_\xi = 0$). **Figure 6—figure supplement 1** further illustrates how the activity varies across the modules as a function of the map size.

The online version of this article includes the following source data and figure supplement(s) for figure 6:

**Source data 1.** Code and data for **Figure 6** and related figure supplements.

**Figure supplement 1.** Transition point in modularity decreases with larger map sizes.

and specificity, aligned with macroscopic gradients of cortical organization. This non-uniformity may play an important functional role supporting feature aggregation (**Hagler and Sereno, 2006**) and the development of mixed representations (**Patel et al., 2014**) in higher (more anterior) cortical areas. Here, we consider two scenarios in the baseline (current-based) model to examine the robustness of our findings to more complex topographic configurations.

First, we varied the size of stimulus-tuned sub-populations (parametrized by $d_i$, see Materials and methods) but kept them fixed across the network. For small sub-populations and intermediate degrees of topographic modularity, the activity along the stimulated pathway decays with network depth, suggesting that input information does not reach the deeper SSNs (see **Figure 6a** and **Figure 6—figure supplement 1**). These results place a lower bound on the size of stimulus-tuned sub-populations below which no signal propagation can occur, as reflected by the negative gain in performance for $d = 0.01$ (**Figure 6b**). Whereas denoising is robust to variation around the baseline value of $d = 0.1$ that yielded perfect partitioning of the feedforward projections (see Supplementary Materials), an upper bound may emerge due to increasing overlap between the maps ($d = 0.2$ in **Figure 6b**). In this case, the activity may 'spill over' to other pathways than the stimulated one, corrupting the input representations and hindering accurate transmission and decoding. This can be alleviated by reduced or no overlap (as in **Figure 6a**), in which case signal propagation and denoising is successful for larger map sizes ($\nu_5^S/\nu_0^S > 1$ also for $d > 0.1$). We thus observe a trade-off between map size, overlap and the degree of topographic precision that is required to accurately propagate stimulus representations (see Discussion).

Second, we took into account the fact that these structural features are known to vary with hierarchical depth resulting in increasingly larger sub-populations and, consequently, increasingly overlapping stimulus selectivity (**Smith et al., 2001**; **Patel et al., 2014**; **Bednar and Wilson, 2016**). To capture this effect, we introduce a linear scaling of map size with depth ($d_{i+1} = \delta + d_i$ for $i \geq 1$, see Materials and methods). The ability of the circuit to gradually clean the signal's representation is fully preserved, as illustrated in **Figure 6c**. In fact, for intermediate modularity ($m < 0.9$) broadening the projections can further sharpen the reconstruction precision (compare curves for $\delta = 0.02$ and $\delta = 0$).

Taken together, these observations demonstrate that a gradual denoising of stimulus inputs can occur entirely as a consequence of the modular wiring between the subsequent processing circuits.

Importantly, this effect generalizes well across diverse neuron and synapse models, as well as key system properties, making modular topography a potentially universal circuit feature for handling noisy data streams.

## Modularity as a bifurcation parameter

The results so far indicate that the modular topographic projections, more so than the individual characteristics of neurons and synapses, lead to a sequential denoising effect through a joint process of signal amplification and noise suppression. To better understand how the system transitions to such an operating regime, it is helpful to examine its macroscopic dynamics in the limit of many sub-networks (*Toyoizumi, 2012*; *Cayco-Gajic and Shea-Brown, 2013*; *Kadmon and Sompolinsky, 2016*). We apply standard mean-field techniques (*Fourcaud and Brunel, 2002*; *Helias et al., 2013*; *Schuecker et al., 2015*) to find the asymptotic firing rates (fixed points across sub-networks) of the stimulated and non-stimulated sub-populations as a function of topography (*Figure 2d*). For this, we can approximate the input $\mu$ to a group of neurons as a linear function of its firing rate $\nu$ with a slope $\kappa$ that is determined by the coupling within the group and an offset given by inputs from other groups of neurons (orange line in *Figure 7a*). With an approximately sigmoidal rate transfer function, the self-consistent solutions are at the intersections marked in *Figure 7a*.

Formally, all neurons in the deep sub-networks of one topographic map form such a group as they share the same firing rate (asymptotic value). The coupling $\kappa$ within this group comprises not only recurrent connections of one sub-network but also modular feedforward projections across sub-networks. For small modularity, the group is in an inhibition-dominated regime ($\kappa < 0$) and we obtain only one fixed point at low activity (*Figure 7a*, left). Importantly, the firing rate of this fixed point is the same for stimulated and non-stimulated topographic maps. Any influence of input signals applied to $SSN_0$ therefore vanishes in the deeper sub-networks and the signal cannot be reconstructed (*fading* regime). As topographic projections become more concentrated (larger $m$), $\kappa$ changes sign and gradually leads to two additional fixed points (as conceptually illustrated in *Figure 7a* and quantified in *Figure 7b* by numerically solving the self-consistent mean-field equations, see also Appendix B): an unstable one (red) that eventually vanishes with increasing $m$ and a stable high-activity fixed point (black). The bistability opens the possibility to distinguish between stimulated and non-stimulated topographic maps and thereby reconstruct the signal in deep sub-networks: in the *active* regime beyond the *critical modularity threshold* (here $m \geq m_{\text{crit}} = 0.76$), a sufficiently strong input signal can drive the activity along the stimulated map to the high-activity fixed point, such that it can permeate the system, while the non-stimulated sub-populations still converge to the low-activity fixed point. Note that this critical modularity represents the minimum modularity value for which bistability emerges. It typically differs from the actual switching point $m_{switch}$, which additionally depends on the input intensity.

In the potential energy landscape $U$ (see Materials and methods), where stable fixed points correspond to minima, the bistability that emerges for more structured topography $m \geq m_{\text{crit}} = 0.76$ can be understood as a transition from a single minimum at low rates (*Figure 7c*, inset) to a second minimum associated with the high-activity state (*Figure 7c*). Even though the full dynamics of the spiking network away from the fixed point cannot be entirely understood in this simplified potential picture (see Appendix B), qualitatively, more strongly modular networks cause deeper potential wells, corresponding to more attractive dynamical states and higher firing rates (see *Figure 9—figure supplement 2*).

Because the intensity of the input signal dictates the rate of different populations in the initial sub-network $SSN_0$ (*Figure 7d*), it also determines, for any given modularity, whether the rate of the stimulated sub-population is in the basin of attraction of the high-activity (see *Figure 7e*, solid markers and arrows) or low-activity (dashed, blue marker and arrow) fixed point. Denoising, and therefore increasing signal reconstruction, is thus achieved by successively (across sub-networks) pushing the population states toward the self-consistent firing rates.

As reported above, for the baseline network and (standard) input ($\lambda = 0.05$) used in *Figures 1 and 2*, the switching point between low and high activity is at $m = 0.83$ (blue markers in *Figure 7d, f*). Stronger input signals move the switching point toward the minimal modularity $m = 0.76$ of the active regime (black markers in *Figure 7d, f*), while weaker inputs only induce a switch at larger modularities (gray markers in *Figure 7d, f*).

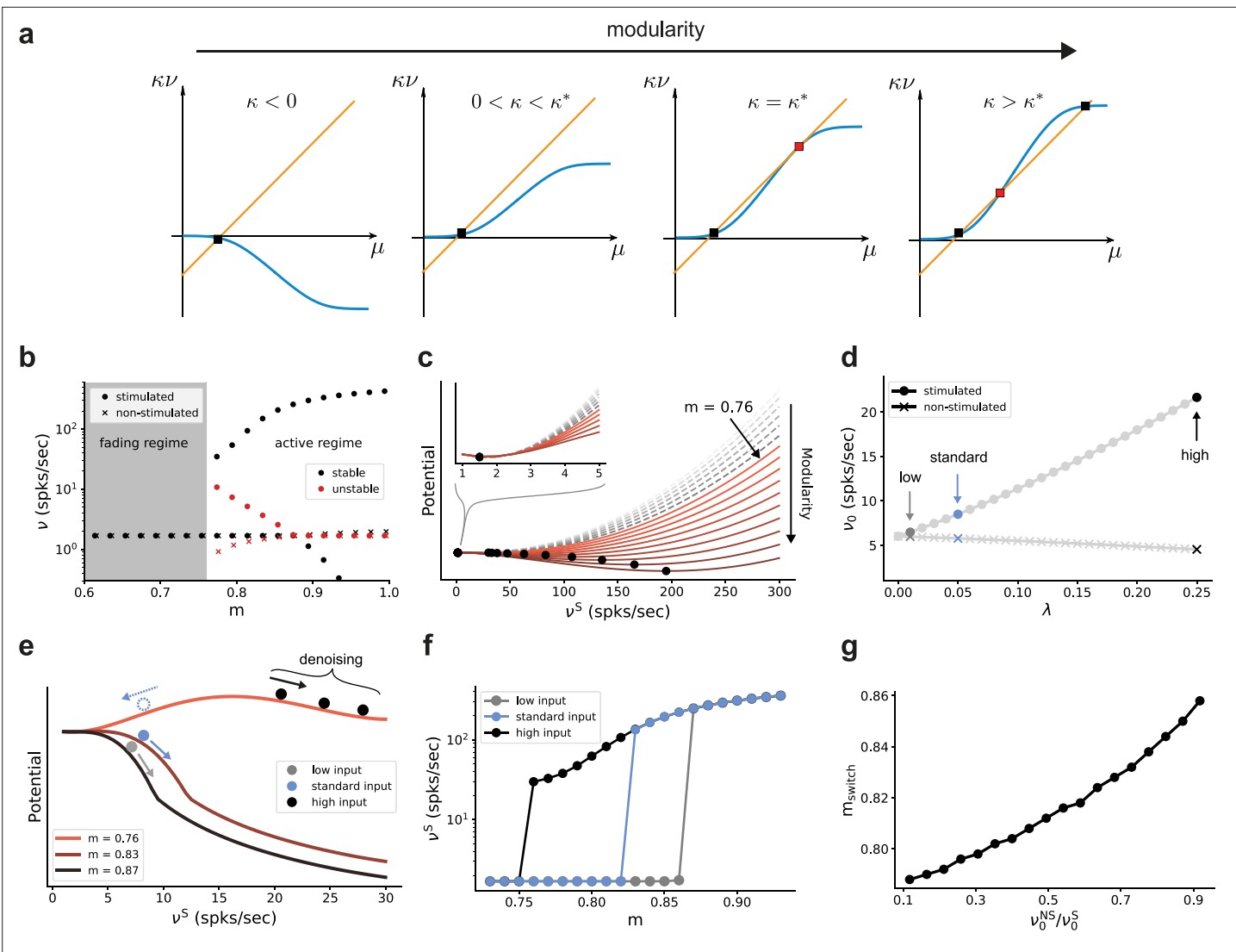

**Figure 7.** Modularity changes the fixed point structure of the system. (**a**) Sketch for self-consistent solution (for the full derivation, see Appendix B) for the firing rate of the stimulated sub-population (blue curves) and the linear relation $\kappa\nu = \mu - I$ (orange lines), in the limit of infinitely deep networks. Squares denote stable (black) and unstable (red) fixed points where input and output rates are the same. (**b**) Bifurcation diagram obtained from numerical evaluation of the mean-field self-consistency equations, *Equations 9 and 10* showing a single stable fixed point in the fading regime, and multiple stable (black) and unstable (red) fixed points in the active regime where denoising occurs. (**c**) Potential energy of the mean activity (see Materials and methods and *Equation 22* in Appendix B) for increasing topographic modularity. A stable state, corresponding to local minimum in the potential, exists at a low non-zero rate in every case, including for $m \leq 0.75$ (gray dashed curves, inset). For $m \geq 0.76$ (colored solid curves), a second fixed point appears at progressively larger firing rates. (**d**) Theoretical predictions for the stationary firing rates of the stimulated and non-stimulated sub-populations in SSN$_0$, as a function of stimulus intensity ($\lambda$, see Materials and methods). Low, standard, and high denote $\lambda$ values of 0.01, 0.05 (baseline value used in *Figure 1*), and 0.25, respectively. (**e**) Sketch of attractor basins in the potential for different values of $m$. Markers correspond to the highlighted initial states in (**d**), with solid and dashed arrows indicating attraction toward the high- and low-activity state, respectively. (**f**) Firing rates of the stimulated sub-population as a function of modularity in the limit of infinite sub-networks, for the three different $\lambda$ marked in (**d**). (**g**) Modularity threshold for the active regime shifts with increasing noise in the input, modeled as additional input to the non-stimulated sub-populations in SSN$_0$. *Figure 7—figure supplement 1* show the dependency of the effective feedforward couplings on different parameters. Note that all panels (except (**a**)) show theoretical predictions obtained from numerical evaluation of the mean-field self-consistency equations.

The online version of this article includes the following source data and figure supplement(s) for figure 7:

**Source data 1.** Code and data for *Figure 7* and related figure supplements.

**Figure supplement 1.** Mean-field predictions for the gain in the firing rates of stimulated sub-populations.

Noise in the input simply shifts the transition point to the high-activity state in a similar manner, with more modular connectivity required to compensate for stronger jitter (**Figure 7g**). However, as long as the mean firing rate of the stimulated sub-population in $SSN_0$ is slightly higher than that of the non-stimulated ones (up to 0.5 spks/sec), it is sufficient to position the system in the attracting basin of the high-rate fixed point and the system is able to clean the signal representation. This indicates a remarkably robust denoising mechanism.

## Critical modularity for denoising

In addition to properties of the input, the critical modularity marking the onset of the active regime is also influenced by neuronal and connectivity features. To build some intuition, it is helpful to consider the sigmoidal activation function of spiking neurons (**Figure 8a**). The nonlinearity of this function prohibits us from obtaining quantitative, closed-form analytical expressions for the critical modularity and requires a numerical solution of the self-consistency equations (**Figure 7b**). However, since the continuous rate model shows a qualitatively similar behavior to the spiking baseline model (see Section 'A generalizable structural effect'), we can study a fully analytically tractable model with piecewise linear activation function (**Figure 8a, b**) to expose the dependence of the critical modularity on both neuron and network properties (see detailed derivations in Appendix B).

In this simple model, the output is zero for inputs below $\mu_{\min} = 15$ and at maximum rate $\nu_{\max} = 150$ for inputs above $\mu_{\max} = 400$. In between these two bounds, the output is linearly interpolated $\nu(\mu) = \nu_{\max}(\mu - \mu_{\min})/(\mu_{\max} - \mu_{\min})$. As discussed before, successful denoising is achieved if the non-stimulated sub-populations are silent, $\nu^{NS} = 0$, and the stimulated sub-populations are active, $\nu^{S} > 0$. Note that in the following we focus on this ideal scenario representing perfect denoising, but, in principle, intermediate solutions with $\nu^{S} \gg \nu^{NS} > 0$ may also occur and could still be considered as successful denoising. Analyzing for which neuron, network and input properties this scenario is achieved, we obtain multiple conditions for the modularity that need to be fulfilled.

The first condition illustrates the dependence of the critical modularity on the neuron model (**Figure 8c**, purple horizontal line)

$$m \geq \frac{(\mu_{\max} - \mu_{\min})N_C}{(1-\alpha)\mathcal{J}\nu_{\max} + (\mu_{\max} - \mu_{\min})(N_C - 1)}, \tag{1}$$

where $N_C$ is the number of stimulus-specific sub-populations and $\alpha \leq 1$ (typically with a value of 0.25) represents the (reduced) noise ratio in the deeper sub-networks, with $\alpha$ scaling the noise and $1 - \alpha$ scaling the feedforward connections (see Materials and methods). This is necessary to ensure that the total excitatory input to each neuron is consistent across the network. In particular, the critical modularity depends on the dynamic range of input $\mu_{\max} - \mu_{\min}$ and output $\nu_{\max}$. The condition represents a lower bound on the modularity required for denoising. Importantly, while it depends on the effective coupling strength $\mathcal{J}$, the noise ratio $\alpha$ and the number of maps $N_C$ (see Materials and methods), it does not depend on the nature of the recurrent interactions (E/I ratio) and the strength of the external background input. In addition, we find two additional critical values of the modularity (cyan and green curves in **Figure 8c–e**), both of which do depend on the strength of the external background input $\nu_X$ and the recurrent connectivity (E/I ratio $\gamma g$):

$$m = \frac{N_C}{N_C - 1} - \frac{1}{N_C - 1}\frac{(1-\alpha)\mathcal{J}\nu_{\max}}{\mu_{\max} - \alpha\mathcal{J}\nu_X - \frac{\mathcal{J}}{N_C}(1+\gamma g)\nu_{\max}} \tag{2}$$

$$m = 1 - \frac{\left(\mu_{\min} - \alpha\mathcal{J}\nu_X - \frac{\mathcal{J}}{N_C}(1+\gamma g)\nu_{\max}\right)}{\mathcal{J}(1-\alpha)\nu_{\max} - (N_C - 1)\left(\mu_{\min} - \alpha\mathcal{J}\nu_X - \frac{\mathcal{J}}{N_C}(1+\gamma g)\nu_{\max}\right)} \tag{3}$$

Depending on the external input strength $\nu_X$, these are either upper or lower bounds. In the denominator of these expressions, the total input (recurrent and external) is compared to the limits of the dynamic range of the neuron model. The cancellation between recurrent and external inputs in the inhibition-dominated baseline model typically yields a total input within the dynamic range of the neuron, such that modularity in feedforward connections can decrease the input of the non-stimulated sub-populations to silence them, and increase the input of the stimulated sub-populations to support their activity. The competition between the excitatory and inhibitory contributions ensures that the total input does not lead to a saturating output activity. Thus, for inhibitory recurrence, denoising can be achieved at a moderate level of modularity over a large range of external background inputs

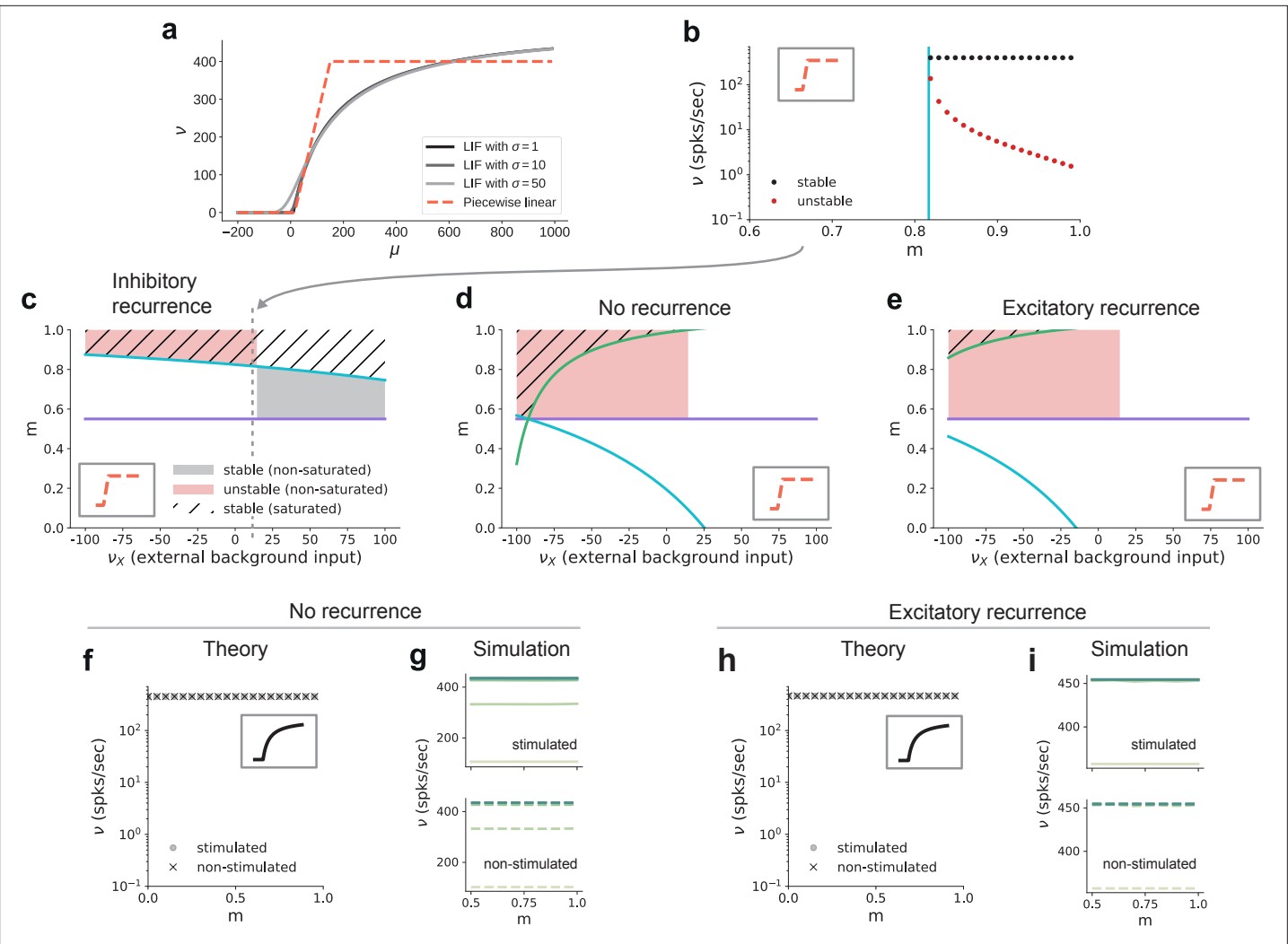

**Figure 8.** Dependence of critical modularity on neuron and connectivity features. (**a**) Activation function $\nu(\mu, \sigma)$ for leaky integrate-and-fire model as a function of the mean input $\mu$ for $\sigma = 1, 10, 50$ (black to gray) and piecewise linear approximation with qualitatively similar shape (red). (**b**) Bifurcation diagram as in *Figure 7b*, but for piecewise linear activation function shown in inset. Low-activity fixed points at zero rate are not shown, which is the case throughout for the non-stimulated sub-populations. This panel corresponds to the cross-section marked by the gray dashed lines in (**c**), at $\nu_X = 12$. Likewise, the vertical cyan bar corresponds to the lower bound on modularity depicted by the cyan curve in (**c**) for the same value $\nu_X = 12$. (**c**) Analytically derived bounds on modularity (purple line corresponds to *Equation 1*, cyan curve to *Equation 2*) as a function of external input for the baseline model with inhibition-dominated recurrent connectivity ($g = -12$). Shaded regions denote positions of stable (black) and unstable (red) fixed points with $0 < \nu^S < \nu_{\max}$ and $\nu^{NS} = 0$. Hatched area represents region with stable fixed points at saturated rates. Denoising occurs in all areas with stable fixed points (hatched and black shaded regions). Negative values on the x-axis correspond to inhibitory external background input with rate $|\nu_X|$. (**d**) Same as panel (**c**) for networks with no recurrent connectivity within the SSNs (green curve defined by *Equation 3*). (**e**) Same as panel (**c**) for networks with excitation-dominated connectivity within SSNs ($g = -3$). (**f**) Same as *Figure 7b*, obtained through numerical evaluation of the mean-field self-consistent equations for the spiking model. All non-zero fixed points are stable, with points representing stimulated (circle) and non-stimulated (cross) populations overlapping. (**g**) Mean firing rates across the SSNs in the current-based (baseline) model with no recurrent connections, obtained from 5 s of network simulations and averaged over five trials. (**h, i**) Same as (**f, g**) for networks with excitation-dominated connectivity.

The online version of this article includes the following source data and figure supplement(s) for figure 8:

**Source data 1.** Code and data for *Figure 8* and related figure supplements.

**Figure supplement 1.** Influence of the activation function's dynamic range on the bifurcation behavior in excitation-dominated networks ($g = -3$, see also *Figure 8e*).

**Figure supplement 2.** Firing rates in $\mathrm{SSN}_5$ in the absence of external background noise ($\nu_X = 0$).

(shaded black and hatched regions in *Figure 8c*), which demonstrates a robust denoising mechanism even in the presence of changes in the input environment.

In contrast, if recurrent connections are absent, strong inhibitory external background input is required to counteract the excitatory feedforward input and achieve a denoising scenario (*Figure 8d*). Fixed points at non-saturated activity $\nu^S > 0$ are also present for low excitatory external input, but unstable due to the positive recurrent feedback. This is because in networks without recurrence, there is no competition between the recurrent input and the external and feedforward inputs. As a result, the input to both the stimulated and non-stimulated sub-populations is typically high, such that modulation of the feedforward input via topography cannot lead to a strong distinction between the pathways as required for denoising. In these networks, one typically observes high activity in all populations. A similar behavior can be observed in excitation-dominated networks (*Figure 8e*), where the inhibitory external background input must be even stronger to compensate the excitatory feedforward and recurrent connectivity and reach a stable denoising regime.

Note that inhibitory external input is not in line with the excitatory nature of external inputs to local circuits in the brain and is therefore biologically implausible. One way to achieve denoising in excitation-dominated networks for excitatory background inputs would be to shift the dynamic range of the activation function (see *Figure 8—figure supplement 1*), which is, however, not consistent with the biophysical properties of real neurons (distance between threshold and rest as compared to typical strengths of postsynaptic potentials). In summary, we find that recurrent inhibition is crucial to achieve denoising in biologically plausible settings.

These results on the role of recurrence and external input can be transferred to the behavior of the spiking model. While details of the fixed point behavior depend on the specific choice of the activation function, *Figure 8f, h* shows that there is also no denoising regime for the spiking model in case of no or excitation-dominated recurrence and a biologically plausible level of external input. Instead, one finds high activity in both stimulated and non-stimulated sub-populations, as confirmed by network simulations (*Figure 8g, i*). *Figure 8—figure supplement 2* further confirms that even reducing the external input to zero does not avoid this high-activity state in both stimulated and non-stimulated sub-populations for $m < 1$.

## Input integration and multi-stability

The analysis considered in the sections above is restricted to a system driven with a single external stimulus. However, to adequately understand the system's dynamics, we need to account for the fact that it can be concurrently driven by multiple input streams. If two simultaneously active stimuli drive the system (see illustration in *Figure 9a*), the qualitative behavior where the responses along the stimulated (non-stimulated) maps are enhanced (silenced) is retained if the strength of the two input channels is sufficiently different (*Figure 9b*, top panel). In this case, the weaker stimulus is not strong enough to drive the sub-population it stimulates toward the basin of attraction of the high-activity fixed point. Consequently, the sub-population driven by this second stimulus behaves as a non-stimulated sub-population and the system remains responsive to only one of the two inputs, acting as a WTA circuit. If, however, the ratio of stimulus intensities varies, two active sub-populations may co-exist (*Figure 9b*, center) and/or compete (bottom panel), depending also on the degree of topographic modularity.

To quantify these variations in macroscopic behavior, we focus on the dynamics of $\text{SSN}_5$ and measure the similarity (correlation coefficient) between the firing rates of the two stimulus-specific sub-populations as a function of modularity and ratio of input intensities $\lambda_2/\lambda_1$ (see Materials and methods and *Figure 9c*). In the case that both inputs have similar intensities but the feedforward projections are not sufficiently modular, both sub-populations are activated simultaneously (Co-Ex, red area in *Figure 9c*). This is the dynamical regime that dominates the earlier sub-networks. However, this is a transient state, and the Co-Ex region gradually shrinks with network depth until it vanishes completely after approximately 9–10 SSNs (see *Figure 9d*).

For low modularity, the system settles in the single stable state associated with near-zero firing rates, as illustrated schematically in the energy landscape in *Figure 9e*, (1) (see Materials and methods, Appendix B, and Supplementary Materials for derivations and numerical simulations). Above the critical modularity value, the system enters one of two different regimes. For $m > 0.84$ and an input ratio below 0.7 (*Figure 9c*, gray area), one stimulus dominates (WTA) and the responses in the two

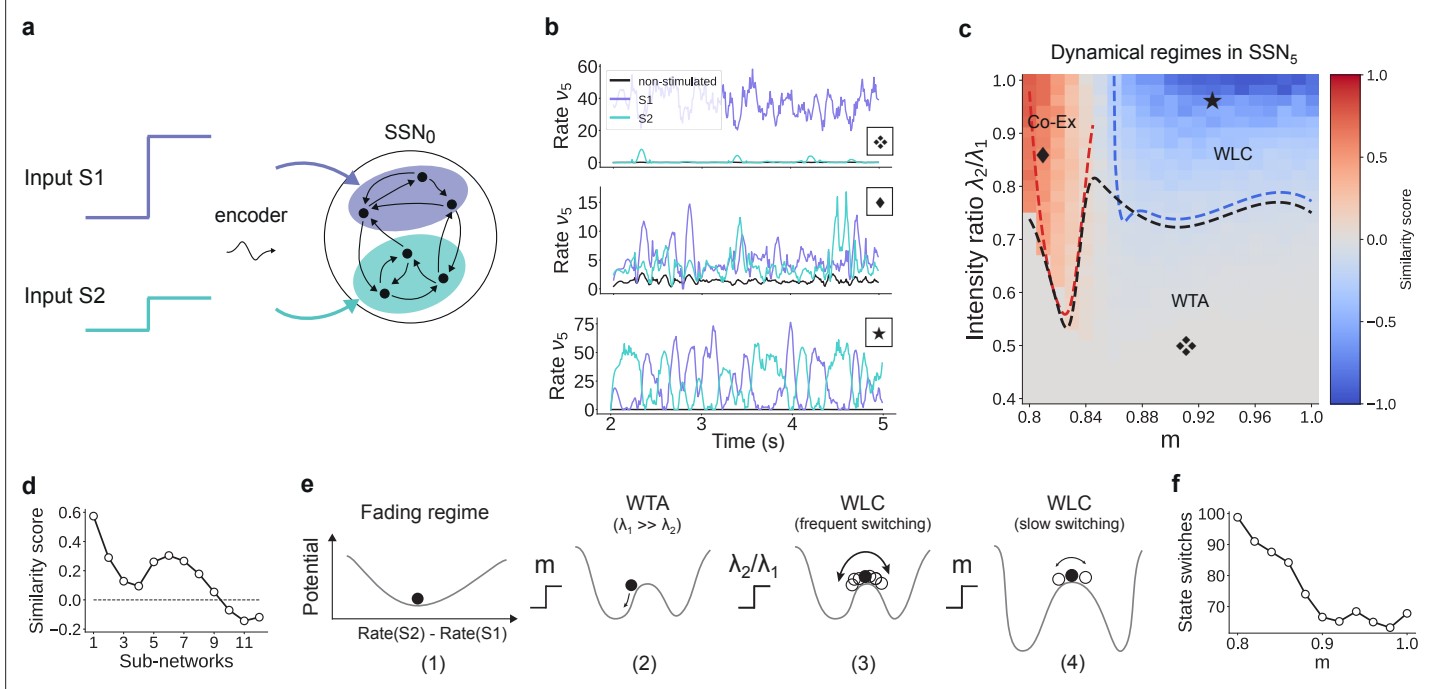

**Figure 9.** For multiple input streams, topography may elicit a wide range of dynamical regimes. (**a**) Two active input channels with corresponding stimulus intensities $\lambda_1$ and $\lambda_2$, mapped onto non-overlapping sub-populations, drive the network simultaneously. Throughout this section, $\lambda_1 = 0.05$ is fixed to the previous baseline value. (**b**) Mean firing rates of the two stimulated sub-populations (purple and cyan), as well as the non-stimulated sub-populations (black) for three different combinations of $m$ and ratios $\lambda_2/\lambda_1$ (as marked in (**c**)). (**c**) Correlation-based similarity score shows three distinct dynamical regimes in $SSN_5$ when considering the firing rates of two, simultaneously stimulated sub-populations associated with $S_1$ and $S_2$, respectively: coexisting (Co-Ex, red area), winner-takes-all (WTA, gray), and winnerless competition (WLC, blue). Curves mark the boundaries between the different regimes (see Materials and methods). Activity for marked parameter combinations shown in (**b**). (**d**) Evolution of the similarity score with increasing network depth, for $m = 0.83$ and input ratio of 0.86. For deep networks, the Co-Ex region vanishes and the system converges to either WLC or WTA dynamics. (**e**) Schematic showing the influence of modularity and input intensity on the system's potential energy landscape (see Materials and methods): (1) in the fading regime there is a single low-activity fixed point (minimum in the potential); (2) increasing modularity creates two high-activity fixed points associated with S1 and S2, with the dynamics always converging to the same minimum due to $\lambda_1 \gg \lambda_2$; (3) strengthening S2 balances the initial conditions, resulting in frequent, fluctuation-driven switching between the two states; (4) for larger $m$ values, switching speed decreases as the wells become deeper and the barrier between the wells wider. (**f**) Switching frequency between the dominating sub-populations in $SSN_5$ decays with increasing modularity. Data computed over 10 s, for $\lambda_2/\lambda_1 = 0.9$. *Figure 9—figure supplement 1* and *Figure 9—figure supplement 2* show the evolution of the Co-Ex region over 12 modules and the potential landscape, respectively.

The online version of this article includes the following source data and figure supplement(s) for figure 9:

**Source data 1.** Code and data for *Figure 9* and related figure supplements.

**Figure supplement 1.** Evolution of similarity score for 12 sub-networks.

**Figure supplement 2.** Potential landscape for two input streams.

populations are uncorrelated (*Figure 9b*, top panel). Although the potential landscape contains two minima corresponding to either population being active, the system always settles in the high-activity attractor state corresponding to the dominating input (*Figure 9e*, (2)).

If, however, the two inputs have comparable intensities and the topographic projections are sharp enough ($m > 0.84$), the system transitions into a different dynamical state where neither stimulus-specific sub-population can maintain an elevated firing rate for extended periods of time. In the extreme case of nearly identical intensities ($\lambda_2/\lambda_1 \geq 0.9$) and high modularity, the responses become anti-correlated (*Figure 9b*, bottom panel), that is the activation of the two stimulus-specific sub-populations switches, as they engage in a dynamic behavior reminiscent of WLC between multiple neuronal groups (*Lagzi and Rotter, 2015*; *Rost et al., 2018*). The switching between the two states is driven by stochastic fluctuations (*Figure 9e*, (3)). The depth of the wells and width of barrier (distance between fixed points) increase with modularity (see *Figure 9e*, (4) and *Figure 9—figure supplement*

2), suggesting a greater difficulty in moving between the two attractors and consequently fewer state changes. Numerical simulations confirm this slowdown in switching (***Figure 9f***).

We wish to emphasize that the different dynamical states arise primarily from the feedforward connectivity profile. Nevertheless, even though the synaptic weights are not directly modified, varying the topographic modularity does translate to a modification of the effective connectivity weights (***Figure 3b***). The ratio of stimulus intensities also plays a role in determining the dynamics, but there is a (narrow) range (approximately between 0.75 and 0.8) for which all 3 regions can be reached through sole modification of the modularity. Together, these results demonstrate that topography can not only lead to spatial denoising but also enable various, functionally important network operating points.

## Reconstruction and denoising of dynamical inputs

Until now, we have considered continuous but piecewise constant, step signals, with each step lasting for a relatively long and fixed period of $200\,\mathrm{ms}$. This may give the impression that the denoising effects we report only works for static or slowly changing inputs, whereas naturalistic stimuli are continuously varying. Nevertheless, sensory perception across modalities relies on varying degrees of temporal and

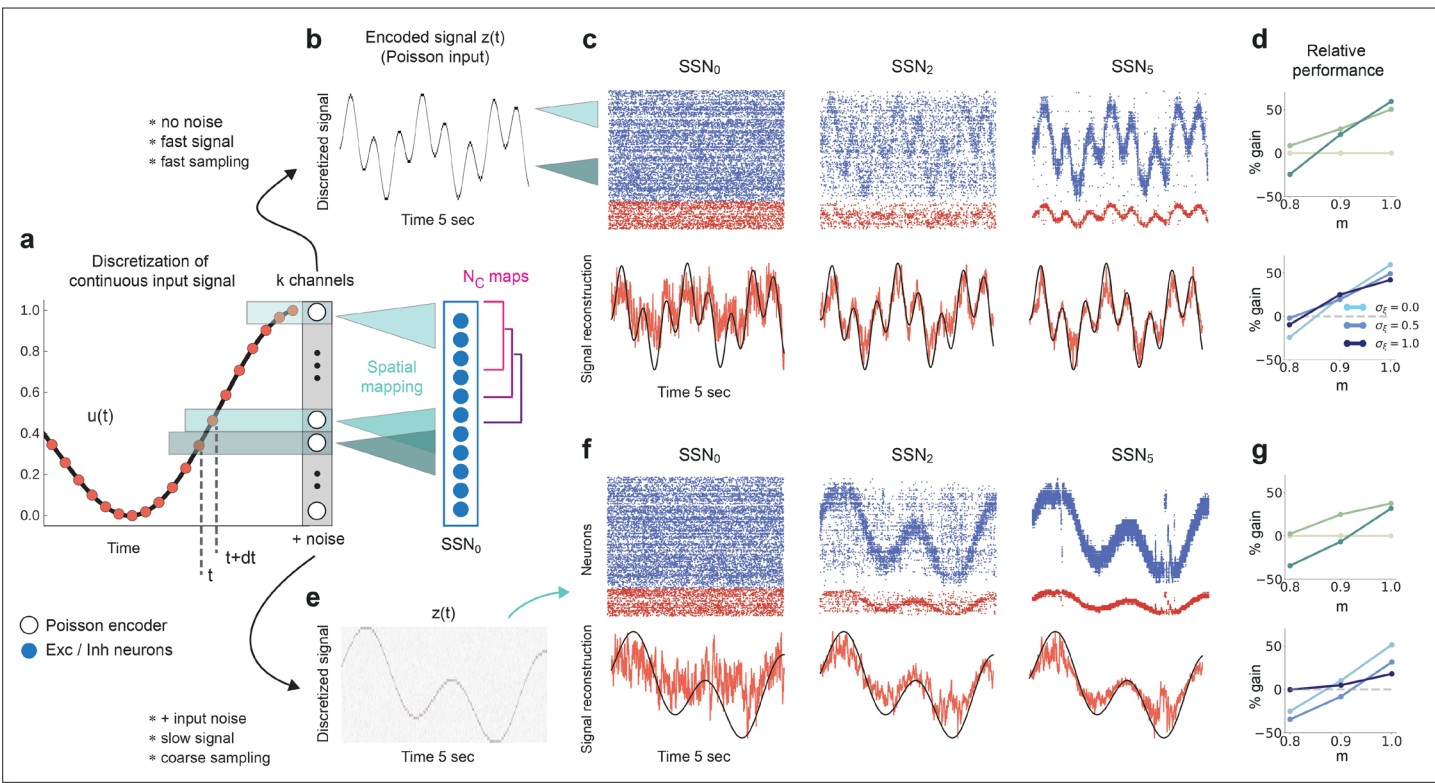

**Figure 10.** Reconstruction of a dynamic, continuous input signal. (**a**) Sketch of the encoding and mapping of a sinusoidal input $x(t)$ onto the current-based network model. The signal is sampled at regular time intervals $dt$, with each sample binned into one of $k$ channels (which is then active for a duration of $dt$). This yields a temporally and spatially discretized $k$-dimensional binary signal $u(t)$, from which we obtain the final noisy input $z(t)$ similar to the baseline network (see ***Figure 1*** and Materials and methods). Unlike the one-to-one mapping in ***Figure 1***, here we decouple the number of channels $k = 40$ from that of topographic maps, $N_C = 20$ (map size is unchanged, $C_i = 800$). Because $N_C < k$, the channels project to evenly spaced but overlapping sub-populations in SSN$_0$, while the maps themselves overlap significantly. (**b**) Discretized signal $z(t)$ and rate encoding for input $x(t) = \sin(10t) + \cos(3t)$, with $dt = 1\,\mathrm{ms}$ and no noise ($\sigma_\xi = 0$). (**c**) Top panel shows the spiking activity of 500 randomly chosen excitatory (blue) and inhibitory (red) neurons in SSN$_0$, SSN$_2$, and SSN$_5$, for $m = 0.9$. Corresponding target signal $x(t)$ (black) and readout output (red) are shown in bottom panel. (**d**) Relative gain in performance in SSN$_2$ and SSN$_5$ for $\sigma_\xi = 0$ (top). Color shade denotes network depth. Bottom panel shows relative gain in SSN$_5$ for different levels of noise $\sigma_\xi \in \{0, 0.5, 1\}$. (**e–g**) Same as (**b–d**), but for a slowly varying signal (sampled at $dt = 20\,\mathrm{ms}$), $\sigma_\xi = 0.5$ and $m = 1$. Performance results are averaged across five trials. We used 20 s of data for training and 10 s for testing (activity sampled every 1 ms, irrespective of input discretization $dt$).

The online version of this article includes the following source data and figure supplement(s) for figure 10:

**Source data 1.** Code and data for ***Figure 10*** and related figure supplements.

**Figure supplement 1.** Limits of denoising for rapidly changing and noisy dynamical inputs.

spatial discretization (*VanRullen and Koch, 2003*), with individual (sub-)features of the input encoded by specific (sub-)populations of neurons in the early stages of the sensory hierarchy. In this section, we will demonstrate that denoising is robust to the temporal properties of the input and its encoding, as we relax many of the assumptions made in previous sections.

We consider a sinusoidal input signal, which we discretize and map onto the network according to the depiction in *Figure 10a*. This approach is similar to previous works, for instance it can mimic the movement of a light spot across the retina (*Klos et al., 2018*). By varying the sampling interval $dt$ and number of channels $k$, we can change the coarseness of the discretization from step-like signals to more continuous approximations of the input. If we choose a high sampling rate ($dt = 1$ ms) and sufficient channels ($k = 40$), we can accurately encode even fast changing signals (*Figure 10b*). Given that each input-driven SSN is inhibition-dominated and therefore close to the balanced state, the network exhibits a fast tracking property (*van Vreeswijk and Sompolinsky, 1996*) and can accurately represent and denoise the underlying continuous signal in the spiking activity (*Figure 10c*, top). This is also captured by the readout, with the tracking precision increasing with network depth (*Figure 10c*, bottom). In this condition, there is a performance gain of up to 50% in the noiseless case (*Figure 10d*, top) and similar values for varying levels of noise (*Figure 10d*, bottom).

Note that due to the increased number of input channels (40 compared to 10) projecting to the same number of neurons in $SSN_0$ as before (800), for the same $\sigma_\xi$ the effective amount of noise each neuron receives is, on average, four times larger than in the baseline network. Moreover, the task was made more difficult by the significant overlap between the maps ($N_C = 20$) as well as the resulting decrease in neuronal input selectivity. Nevertheless, similar results were obtained for slower and more coarsely sampled signals (*Figure 10e–g*).

We found comparable denoising dynamics for a large range of parameter combinations involving the map size, number of maps, number of channels, and signal complexity. Although there are limits with respect to the frequencies (and noise intensity) the network can track (see *Figure 10—figure supplement 1*), these findings indicate a very robust and flexible phenomenon for denoising spatially encoded sensory stimuli.

## Discussion

The presence of stimulus- or feature-tuned sub-populations of neurons in primary sensory cortices (as well as in downstream areas) provides an efficient spatial encoding strategy (*Pouget et al., 1999*; *Seriès et al., 2004*; *Tkacik et al., 2010*) that ensures the relevant computable features are accurately represented. Here, we propose that beyond primary sensory areas, modular topographic projections play a key role in preserving accurate representations of sensory inputs across many processing modules. Acting as a structural scaffold for a sequential denoising mechanism, we show how they simultaneously enhance relevant stimulus features and remove noisy interference. We demonstrate this phenomenon in a variety of network models and provide a theoretical analysis that indicates its robustness and generality.

When reconstructing a spatially encoded input signal corrupted by noise in a network of sequentially connected populations, we find that a convergent structure in the feedforward projections is not only critical for successfully solving the task, but that the performance increases significantly with network depth beyond a certain modularity (*Figure 1*). Through this mechanism, the response selectivity of the stimulated sub-populations is sharpened within each subsequent sub-network, while others are silenced (*Figure 2*). Such wiring may support efficient and robust information transmission from the thalamus to deeper cortical centers, retaining faithful representations even in the presence of strong noise. We demonstrate that this holds for a variety of signals, from approximately static (stepwise) to smoothly and rapidly changing dynamic inputs (*Figure 10*). Thanks to the balance of excitation and inhibition, the network is able to track spatially encoded signals on very short timescales, and is flexible with respect to the level of spatial and temporal discretization. Accurate tracking and denoising requires that the encoding is locally static/semi-stationary for only a few tens of milliseconds, which is roughly in line with psychophysics studies on the limits of sensory perception (*Borghuis et al., 2019*).

More generally, topographic modularity, in conjunction with other top-down processes (*Kok et al., 2012*), could provide the anatomical substrate for the implementation of a number of behaviorally relevant processes. For example, feedforward topographic projections on the visual pathway could contribute, together with various attentional control processes, to the widely observed *pop-out effect*

in the later stages of the visual hierarchy (*Brefczynski-Lewis et al., 2009*; *Itti et al., 1998*). The pop-out effect, at its core, assumes that in a given context some neurons exhibit sharper selectivity to their preferred stimulus feature than the neighboring regions, which can be achieved through a winner-take-all (WTA) mechanism (see *Figure 9* and *Himberger et al., 2018*).

The WTA behavior underlying the denoising is caused by a re-shaping of the E/I balance across the network (see *Figure 3*). As the excitatory feedforward projections become more focused, they modulate the system's effective connectivity and thereby the gain on the stimulus-specific pathways, gating or allowing (and even enhancing) signal propagation. This change renders the stimulated pathway excitatory in the active regime (see *Figure 7*), leading to multiple fixed points such as those observed in networks with local recurrent excitation (*Renart et al., 2007*; *Litwin-Kumar and Doiron, 2012*). While the high-activity fixed point of such clustered networks is reached over time, in our model it unfolds progressively in space, across multiple populations. Importantly, in the range of biologically plausible numbers of cortical areas relevant for signal transmission (up to 10 for some visual stimuli, see *Felleman and Van Essen, 1991*; *Hegdé and Felleman, 2007*) and intermediate modularity, the firing rates remain within experimentally observed limits and do not saturate. The basic principle is similar to other approaches that alter the gain on specific pathways to facilitate stimulus propagation, for example through stronger synaptic weights (*Vogels and Abbott, 2005*), stronger nonlinearity (*Toyoizumi, 2012*), tuning of connectivity strength, and neuronal thresholds (*Cayco-Gajic and Shea-Brown, 2013*), via detailed balance of local excitation and inhibition (amplitude gating; *Vogels and Abbott, 2009*) or with additional subcortical structures (*Cortes and van Vreeswijk, 2015*). Additionally, our model also displays some activity characteristics reported previously, such as the response sharpening observed for synfire chains (*Diesmann et al., 1999*) or (almost) linear firing rate propagation (*Kumar et al., 2010*) (for intermediate modularity).

However, due to the reliance on increasing inhibitory activity at every stage, we speculate that denoising, as studied here, would not occur in such a system containing a single, shared inhibitory pool with homogeneous connectivity. In this case, inhibition would affect all excitatory populations uniformly, with stronger activity potentially preventing accurate stimulus transmission from the initial sub-networks. Nevertheless, this problem could be alleviated using a more realistic, localized spatial connectivity profile as in *Kumar et al., 2008a*, or by adding shadow pools (groups of inhibitory neurons) for each layer of the network, carefully wired in a recurrent or feedforward manner (*Aviel et al., 2003*; *Aviel et al., 2005*; *Vogels and Abbott, 2009*). In such networks with non-random or spatially dependent connectivity, structured (modular) topographic projections onto the inhibitory populations will likely be necessary to maintain stable dynamics and attain the appropriate inhibition-dominated regimes (*Figure 3*). Alternatively, these could be achieved through additional, targeted inputs from other areas (*Figure 4*), with feedforward inhibition known to provide a possible mechanism for context-dependent gating or selective enhancement of certain stimulus features (*Ferrante et al., 2009*; *Roberts et al., 2013*).

While our findings build on the above results, we here show that the experimentally observed topographic maps may serve as a structural denoising mechanism for sensory stimuli. In contrast to most works on signal propagation where noise mainly serves to stabilize the dynamics and is typically avoided in the input, here the system is driven by a continuous signal severely corrupted by noise. Taking a more functional approach, this input is reconstructed using linear combinations of the full network responses, rather than evaluating the correlation structure of the activity or relying on precise firing rates. Focusing on the modularity of such maps in recurrent spiking networks, our model also differs from previous studies exploring optimal connectivity profiles for minimizing information loss in purely feedforward networks (*Renart and van Rossum, 2012*; *Zylberberg et al., 2017*), also in the context of sequential denoising autoencoders (*Kadmon and Sompolinsky, 2016*) and stimulus classification (*Babadi and Sompolinsky, 2014*), which used simplified neuron models or shallow networks, made no distinction between excitatory and inhibitory connections, or relied on specific, trained connection patterns (e.g., chosen by the pseudo-inverse model). Although the bistability underlying denoising can, in principle, also be achieved in such feedforward or networks without inhibition, our theoretical predictions and network simulations indicate that for biologically constrained circuits (i.e., where the background and long-range feedforward input is excitatory), inhibitory recurrence is indispensable for the spatial denoising studied here (see Section 'Critical modularity for denoising').

Recurrent inhibition compensates for the feedforward and external excitation, generating competition between the topographic pathways and allowing the populations to rapidly track their input.

Moreover, our findings provide an explanation for how low-intensity stimuli (1–2 spks/sec above background activity, see *Figure 2* and Supplementary Materials) could be amplified across the cortex despite significant noise corruption, and relies on a generic principle that persists across different network models (*Figure 5*) while also being robust to variations in the map size (*Figure 6*). We demonstrated both the existence of a lower and upper (due to increased overlap) bound on their spatial extent for signal transmission, as well as an optimal region for which denoising was most pronounced. These results indicate a trade-off between modularity and map size, with larger maps sustaining stimulus propagation at lower modularity values, whereas smaller maps must compensate through increased topographic density (see *Figure 6a* and Supplementary Materials). In the case of smaller maps, progressively enlarging the receptive fields enhanced the denoising effect and improved task performance (*Figure 6c*), suggesting a functional benefit for the anatomically observed decrease in topographic specificity with hierarchical depth (*Bednar and Wilson, 2016*; *Smith et al., 2001*). One advantage of such a wiring could be spatial efficiency in the initial stages of the sensory hierarchy due to anatomical constraints, for instance the retina or the lateral geniculate nucleus. While we get a good qualitative description of how the spatial variation of topographic maps influences the system's computational properties, the numerical values in general are not necessarily representative. Cortical maps are highly dynamic and exhibit more complex patterning, making (currently scarce) precise anatomical data a prerequisite for more detailed investigations. For instance, despite abundant information on the size of receptive fields (*Smith et al., 2001*; *Liu et al., 2016*; *Keliris et al., 2019*), there is relatively little data on the connectivity between neurons tuned to related or different stimulus features across distinct cortical circuits. Should such experiments become feasible in the future, our model provides a testable prediction: the projections must be denser (or stronger) between smaller maps to allow robust communication whereas for larger maps fewer connections may be sufficient.

Finally, our model relates topographic connectivity to competition-based network dynamics. For two input signals of comparable intensities, moderately structured projections allow both representations to coexist in a decodable manner up to a certain network depth, whereas strongly modular connections elicit WLC like behavior characterized by stochastic switching between the two stimuli (see *Figure 9*). Computation by switching is a functionally relevant principle (*McCormick, 2005*; *Schittler Neves and Timme, 2012*), which relies on fluctuation- or input-driven competition between different metastable (unstable) or stable attractor states. In the model studied here, modular topography induced multi-stability (uncertainty) in representations, alternating between two stable fixed points corresponding to the two input signals. Structured projections may thus partially explain the experimentally observed competition between multiple stimulus representations across the visual pathway (*Li et al., 2016*), and is conceptually similar to an attractor-based model of perceptual bistability (*Moreno-Bote et al., 2007*). Moreover, this multi-stability across sub-networks can be 'exploited' at any stage by control signals, that is additional modulation (inihibitory) could suppress one and amplify (bias) another.

Importantly, all these different dynamical regimes emerge progressively through the hierarchy and are not discernible in the initial modules. Previous studies reporting on similar dynamical states have usually considered either the synaptic weights as the main control parameter (*Lagzi and Rotter, 2015*; *Lagzi et al., 2019*; *Vogels and Abbott, 2005*) or studied specific architectures with clustered connectivity (*Schaub et al., 2015*; *Litwin-Kumar and Doiron, 2012*; *Rost et al., 2018*). Our findings suggest that in a hierarchical circuit a similar palette of behaviors can be also obtained given appropriate effective connectivity patterns modulated exclusively through modular topography. Although we used fixed projections throughout this study, these could also be learned and shaped continuously through various forms of synaptic plasticity (see e.g. *Tomasello et al., 2018*). To achieve such a variety of dynamics, cortical circuits most likely rely on a combination of all these mechanisms, that is, pre-wired modular connections (within and between distant modules) and heterogeneous gain adaptation through plasticity, along with more complex processes such as targeted inhibitory gating.

Overall, our results highlight a novel functional role for topographically structured projection pathways in constructing reliable representations from noisy sensory signals, and accurately routing them across the cortical circuitry despite the plethora of noise sources along each processing stage.

## Materials and methods

### Network architecture

We consider a feedforward network architecture where each sub-network (SSN) is a balanced random network (**Brunel, 2000**) composed of $N = 10000$ homogeneous LIF neurons, grouped into a population of $N^E = 0.8N$ excitatory and $N^I = 0.2N$ inhibitory units. Within each sub-network, neurons are connected randomly and sparsely, with a fixed number of $K_E = \epsilon N^E$ local excitatory and $K_I = \epsilon N^I$ local inhibitory inputs per neuron. The sub-networks are arranged sequentially, that is the excitatory neurons $E_i$ in $SSN_i$ project to both $E_{i+1}$ and $I_{i+1}$ populations in the subsequent sub-network $SSN_{i+1}$ (for an illustrative example, see **Figure 1a**). There are no inhibitory feedforward projections. Although projections between sub-networks have a specific, non-uniform structure (see next section), each neuron in $SSN_{i+1}$ receives the same total number of synapses from the previous SSN, $K_{FF}$.

In addition, all neurons receive $K_X$ inputs from an external source representing stochastic background noise. For the first sub-network, we set $K_X = K_E$, as it is commonly assumed that the number of background input synapses modeling local and distant cortical input is in the same range as the number of recurrent excitatory connections (see e.g. **Brunel, 2000**; **Kumar et al., 2008b**; **Duarte and Morrison, 2014**). To ensure that the total excitatory input to each neuron is consistent across the network, we scale $K_X$ by a factor of $\alpha = 0.25$ for the deeper SSNs and set $K_{FF} = (1 - \alpha)K_E$, resulting in a ratio of 3:1 between the number of feedforward and background synapses.

### Modular feedforward projections

Within each SSN, each neuron is assigned to one or more of $N_C$ sub-populations *SP* associated with a specific stimulus ($N_C = 10$ unless otherwise stated). This is illustrated in **Figure 1a** for $N_C = 2$. We choose these sub-populations so as to minimize their overlap within each $SSN_i$, and control their effective size $C_i^\beta = d_i N^\beta, \beta \in [E, I]$, through the scaling parameter $d_i \in [0, 1]$. Depending on the size and number of sub-populations, it is possible that some neurons are not part of any or that some neurons belong to multiple such sub-populations (overlap).

### Map size

In what follows, a topographic map refers to the sequence of sub-populations in the different sub-networks associated with the same stimulus. To enable a flexible manipulation of the map sizes, we constrain the scaling factor $d_i$ by introducing a step-wise linear increment $\delta$, such that $d_i = d_0 + i\delta, i \geq 1$. Unless otherwise stated, we set $d_0 = 0.1$ and $\delta = 0$. Note that all SPs within a given SSN have the same size. In this study, we will only explore values in the range $0 \leq \delta \leq 0.02$ to ensure consistent map sizes across the system, that is, $0 \leq d_i \leq 1$ for all $SSN_i$ (see constraints in Appendix A).

### Modularity

To systematically modify the degree of modular segregation in the topographic projections, we define a modularity parameter that determines the relative probability for feedforward connections from a given SP in $SSN_i$ to target the corresponding SP in $SSN_{i+1}$. Specifically, we follow (**Newman, 2009**; **Pradhan et al., 2011**) and define $m = 1 - \frac{p_0}{p_c} \in [0, 1]$ as the ratio of the feedforward projection probabilities between neurons belonging to different SPs ($p_0$) and between neurons on the same topographic map ($p_c$). According to the above definition, the feedforward connectivity matrix is random and homogeneous (Erdős-Rényi graph) if $m = 0$ or $d_i = 1$ (see **Figure 1a**). For $m = 1$ it is a block-diagonal matrix, where the individual SPs overlap only when $d_i > 1/N_C$. In order to isolate the effects on the network dynamics and computational performance attributable exclusively to the topographic structure, the overall density of the feedforward connectivity matrix is kept constant at $(1 - \alpha) * \epsilon = 0.075$ (see also previous section). We note that, while providing the flexibility to implement the variations studied in this manuscript, this formalism has limitations (see Appendix A).

### Neuron and synapse model

We study networks composed of LIF neurons with fixed voltage threshold and static synapses with exponentially decaying postsynaptic currents or conductances. The sub-threshold membrane potential dynamics of such a neuron evolves according to:

$$\tau_{\mathrm{m}} \frac{dV(t)}{dt} = \left(V_{\mathrm{rest}} - V(t)\right) + R\left(I^{\mathrm{E}}(t) + I^{\mathrm{I}}(t) + I^{\mathrm{X}}(t)\right) \tag{4}$$

where $\tau_{\mathrm{m}}$ is the membrane time constant, and $RI^{\beta}$ is the total synaptic input from population $\beta \in [E, I]$. The background input $I^{\mathrm{X}}$ is assumed to be excitatory and stochastic, modeled as a homogeneous Poisson process with constant rate $\nu_{\mathrm{X}}$. Synaptic weights $J_{\mathrm{ij}}$, representing the efficacy of interaction from presynaptic neuron $j$ to postsynaptic neuron $i$, are equal for all realized connections of a given type, that is, $J_{\mathrm{EE}} = J_{\mathrm{IE}} = J$ for excitatory and $J_{\mathrm{EI}} = J_{\mathrm{II}} = gJ$ for inhibitory synapses. All synaptic delays and time constants are equal in this setup. For a complete, tabular description of the models and model parameters used throughout this study, see *Supplementary files 1–5*.

Following previous works (*Zajzon et al., 2019*; *Duarte and Morrison, 2014*), we choose the intensity of the stochastic input $\nu_{\mathrm{X}}$ and the E–I ratio $g$ such that the first two sub-networks operate in a balanced, asynchronous irregular regime when driven solely by background input. This is achieved with $\nu_{\mathrm{X}} = 12$ spikes/s and $g = -12$, resulting in average firing rates of $\sim 3$ spikes/s, coefficient of variation ($CV_{\mathrm{ISI}}$) in the interval $[1.0, 1.5]$ and Pearson cross-correlation (CC) $\leq 0.01$ in $\mathrm{SSN}_0$ and $\mathrm{SSN}_1$.

In Section 'A generalizable structural effect' we consider two additional systems, a network of LIF neurons with conductance-based synapses and a continuous firing rate model. The LIF network is described in detail in *Zajzon et al., 2019*. Spike-triggered synaptic conductances are modeled as exponential functions, with fixed and equal conduction delays for all synapses. Key differences to the current-based model include, in addition to the biologically more plausible synapse model, longer synaptic time constants and stronger input (see also *Zajzon et al., 2019* and *Supplementary file 3* for the numerical values of all parameters).

The continuous rate model contains $N = 3000$ nonlinear units, the dynamics of which are governed by:

$$\tau_x \frac{d\boldsymbol{x}}{dt} = -\boldsymbol{x} + J\boldsymbol{r} + J^{\mathrm{in}}\boldsymbol{u} - \boldsymbol{b}^{\mathrm{rec}} + \sqrt{2\tau_{\boldsymbol{x}}}\sigma_{\mathrm{X}}\boldsymbol{\xi}$$

$$\boldsymbol{r} = 0.5(1 + \tanh\left(\boldsymbol{x}\right)) \tag{5}$$

where $\boldsymbol{x}$ represents the activation and $\boldsymbol{r}$ the output of all units, commonly interpreted as the synaptic current variable and the firing rate estimate, respectively. The rates $r_{\mathrm{i}}$ are obtained by applying the nonlinear transfer function $\tanh(x_{\mathrm{i}})$, modified here to constrain the rates to the interval $[0, 1]$ is the neuronal time constant, $\boldsymbol{b}^{\mathrm{rec}}$ is a vector of individual neuronal bias terms (i.e., a baseline activation), and $J$ and $J^{\mathrm{in}}$ are the recurrent (including feedforward) and input weight matrices, respectively. These are constructed in the same manner as for the spiking networks, such that the overall connectivity, including the input mapping onto $\mathrm{SSN}_0$, is identical for all three models. Input weights are drawn from a uniform distribution, while the rest follow a normal distribution. Finally, $\boldsymbol{\xi}$ is a vector of $N$ independent realizations of Gaussian white noise with zero mean and variance scaled by $\sigma_{\mathrm{X}}$. The differential equations are integrated numerically, using the Euler–Maruyama method with step $\delta t = 1$ ms, with specific parameter values given in *Supplementary file 5*.

## Signal reconstruction task

We evaluate the system's ability to recover a simple, continuous step signal from a noisy variation using linear combinations of the population responses in the different SSNs (*Maass et al., 2002*). This is equivalent to probing the network's ability to function as a denoising autoencoder (*Bengio et al., 2013*).

To generate the $N_{\mathrm{C}}$-dimensional input signal $u(t)$, we randomly draw stimuli from a predefined set $S = \{S_1, S_2, ..., S_{N_C}\}$ and set the corresponding channel to active for a fixed duration of 200 ms (*Figure 1a*, left). This binary step signal $u(t)$ is also the target signal to be reconstructed. The effective input is obtained by adding a Gaussian white noise process with zero mean and variance $\sigma_{\xi}^2$ to $u(t)$, and scaling the sum with the input rate $\nu_{\mathrm{in}}$. Rectifying the resulting signal leads to the final form of the continuous input signal $z(t) = [\nu_{\mathrm{in}}(u(t) + \xi(t))]_+$. This allows us to control the amount of noise in the input, and thus the task difficulty, through a single parameter $\sigma_{\xi}$.

To deliver the input to the circuit, the analog signal $z(t)$ is converted into spike trains, with its amplitude serving as the rate of an inhomogeneous Poisson process generating independent spike trains. We set the scaling amplitude to $\nu_{\mathrm{in}} = K_E\lambda\nu_{\mathrm{X}}$, modeling stochastic input with fixed rate $\lambda\nu_{\mathrm{X}}$ from

$K_E = 800$ neurons. If not otherwise specified, $\lambda = 0.05$ holds, resulting in a mean firing rate below 8 spks/sec in $\text{SSN}_0$ (see *Figure 2c*).

Each input channel $k$ is mapped onto one of the $N_C$ stimulus-specific sub-populations of excitatory and inhibitory neurons in the first (input) sub-network $\text{SSN}_0$, chosen according to the procedure described above (see also *Figure 1a*). This way, each stimulus $S_k$ is mapped onto a specific set of sub-populations in the different sub-networks, that is, the topographic map associated with $S_k$.

For each stimulus in the sequence, we sample the responses of the excitatory population in each $\text{SSN}_i$ at fixed time points (once every ms) relative to stimulus onset. We record from the membrane potentials $V_m$ as they represent a parameter-free and direct measure of the population state (*Duarte et al., 2018*; *Uhlmann et al., 2017*). The activity vectors are then gathered in a state matrix $X_{\text{SSN}_i} \in \mathbb{R}^{N^E \times T}$, which is then used to train a linear readout to approximate the target output of the task (*Lukoševičius and Jaeger, 2009*). We divide the input data, containing a total of 100 stimulus presentations (yielding $T = 20{,}000$ samples), into a training and a testing set (80/20%), and perform the training using ridge regression (L2 regularization), with the regularization parameter chosen by leave-one-out cross-validation on the training dataset.

Reconstruction performance is measured using the normalized root mean squared error (NRMSE). For this particular task, the effective delay in the build-up of optimal stimulus representations varies greatly across the sub-networks. In order to close in on the optimal delay for each $\text{SSN}_i$, we train the state matrix $X_{\text{SSN}_i}$ on a larger interval of delays and choose the one that minimizes the error, averaged across multiple trials.

In Section 'Reconstruction and denoising of dynamical inputs', we generalize the input to a sinusoidal signal $x(t) = \sin(a \cdot t) + \cos(b \cdot t)$, with parameters $a$ and $b$. From this, we obtain $u(t)$ through the sampling and discretization process described in the respective section, and compute the final input $z(t) = [\nu_{\text{in}}(u(t) + \xi(t))]_+$ as above.

## Effective connectivity and stability analysis

To better understand the role of structural variations on the network's dynamics, we determine the network's effective connectivity matrix $W$ analytically by linear stability analysis around the system's stationary working points (see Appendix B for the complete derivations). The elements $w_{ij} \in W$ represent the integrated linear response of a target neuron $i$, with stationary rate $\nu_i$, to a small perturbation in the input rate $\nu_j$ caused by a spike from presynaptic neuron $j$. In other words, $w_{ij}$ measures the average number of additional spikes emitted by a target neuron $i$ in response to a spike from the presynaptic neuron $j$, and its relation to the synaptic weights is defined by *Tetzlaff et al., 2012*; *Helias et al., 2013*:

$$
\begin{aligned}
w_{ij} &= \frac{\partial \nu_i}{\partial \nu_j} = \widetilde{\alpha} J_{ij} + \widetilde{\beta} J_{ij}^2 \\
\text{with} \quad \widetilde{\alpha} &= \sqrt{\pi} \left(\tau_m \nu_i\right)^2 \frac{1}{\sigma_i} \left(f(y_\theta) - f(y_r)\right) \\
\text{and} \quad \widetilde{\beta} &= \sqrt{\pi} \left(\tau_m \nu_i\right)^2 \frac{1}{2\sigma_i^2} \left(f(y_\theta)y_\theta - f(y_r)y_r\right) .
\end{aligned}
\tag{6}
$$

Note that in *Figure 3* we ignore the contribution $\widetilde{\beta}$ resulting from the modulation in the input variance $\sigma_j^2$ which is significantly smaller due to the additional factor $1/\sigma_i \sim \mathcal{O}(1/\sqrt{N})$. Importantly, the effective connectivity matrix $W$ allows us to gain insights into the stability of the system by eigenvalue decomposition. For large random coupling matrices, the effective weight matrix has a spectral radius $\rho = \max_k \left(\text{Re}\{\lambda_k\}\right)$ which is determined by the variances of $W$ (*Rajan and Abbott, 2006*). For inhibition-dominated systems, such as those we consider, there is a single negative outlier representing the mean effective weight, given the eigenvalue $\lambda_k^*$ associated with the unit vector. The stability of the system is thus uniquely determined by the spectral radius $\rho$: values smaller than unity indicate stable dynamics, whereas $\rho > 1$ lead to unstable linearized dynamics.

## Fixed point analysis

For the mean-field analysis, the $N_C$ sub-populations in each sub-network can be reduced to only two groups of neurons, the first one comprising all neurons of the stimulated SPs and the second one comprising all neurons in all non-stimulated SPs. This is possible because (1) the firing rates of the excitatory and inhibitory neurons within one SP are identical, owing to homogeneous neuron parameters and matching incoming connection statistics, and (2) all neurons in non-stimulated SPs have the

same rate $\nu^{\mathrm{NS}}$ that is in general different from the rate of the stimulated SP $\nu^{\mathrm{S}}$. Here we only sketch the main steps, with a detailed derivation given in Appendix B.

The mean inputs to the first sub-network can be obtained via

$$
\begin{aligned}
\mu^{\mathrm{S}} &= (1+\lambda)\mathcal{J}\nu_{\mathrm{x}} + \tfrac{1}{N_{\mathrm{C}}}\mathcal{J}\left(1+\gamma g\right)\nu^{\mathrm{S}} + \tfrac{N_{\mathrm{C}}-1}{N_{\mathrm{C}}}\mathcal{J}\left(1+\gamma g\right)\nu^{\mathrm{NS}}, \\
\mu^{\mathrm{NS}} &= \mathcal{J}\nu_{\mathrm{x}} + \tfrac{1}{N_{\mathrm{C}}}\mathcal{J}\left(1+\gamma g\right)\nu^{\mathrm{S}} + \tfrac{N_{\mathrm{C}}-1}{N_{\mathrm{C}}}\mathcal{J}\left(1+\gamma g\right)\nu^{\mathrm{NS}}
\end{aligned}
\tag{7}
$$

where $\gamma = K_{\mathrm{I}}/K_{\mathrm{E}}$ and $\mathcal{J} = \tau_{\mathrm{m}}K_{\mathrm{E}}J$. Both equations are of the form

$$
\kappa\nu = \mu - I
\tag{8}
$$

where $\kappa$ is the effective self-coupling of a group of neurons with rate $\nu$ and input $\mu$, and $I$ denotes the external inputs from other groups. *Equation 8* describes a linear relationship between the rate $\nu$ and the input $\mu$. To find a self-consistent solution for the rates $\nu^{\mathrm{S}}$ and $\nu^{\mathrm{NS}}$, the above equations need to be solved numerically, taking into account in addition the f–I curve $\nu(\mu)$ of the neurons that in the case of LIF model neurons also depends on the variance $\sigma^2$ of inputs. The latter can be obtained analogous to the mean input $\mu$ (see Appendix B). Note that for general nonlinearity $\nu(\mu)$ there is no analytical closed-form solution for the fixed points.

Starting from $\mathrm{SSN}_1$, networks are connected in a fixed pattern such that the rate $\nu_i$ in $\mathrm{SSN}_i$ also depends on the excitatory input from the previous sub-network $\mathrm{SSN}_{i-1}$ with rate $\nu_{i-1}$. For a fixed point, we have $\nu_i = \nu_{i-1}$ (*Toyoizumi, 2012*). In this case, we can effectively group together stimulated/non-stimulated neurons in successive sub-networks and re-group equations for the mean input in the limit of many sub-networks, obtaining the simplified description (details see Appendix B)

$$
\mu^{\mathrm{S}} = \alpha\mathcal{J}\nu_{\mathrm{x}} + \kappa_{\mathrm{S,S}}\,\nu^{\mathrm{S}} + \kappa_{\mathrm{S,NS}}\,\nu^{\mathrm{NS}}
\tag{9}
$$

$$
\mu^{\mathrm{NS}} = \alpha\mathcal{J}\nu_{\mathrm{x}} + \kappa_{\mathrm{NS,S}}\,\nu^{\mathrm{S}} + \kappa_{\mathrm{NS,NS}}\,\nu^{\mathrm{NS}}
\tag{10}
$$

The scaling terms of the firing rates incorporate the recurrent and feedforward contributions from the stimulated and non-stimulated groups of neurons. They depend solely on some fixed parameters of the system, including modularity $m$ (see Appendix B). Importantly, *Equations 9 and 10* and have the same linear form as (*Equation 8*) *Equation 8* and can be solved numerically as described above. Again, for general nonlinear $\nu(\mu)$ there is no closed-form analytical solution, but see below for a piecewise linear activation function $\nu(\mu)$. The numerical solutions for fixed points are obtained using the root finding algorithm root of the scipy.optimize package (*Virtanen et al., 2020*). The stability of the fixed points is obtained by inserting the corresponding firing rates into the effective connectivity *Equation 6*. On the level of stimulated and non-stimulated sub-populations, the effective connectivity matrix reads

$$
\frac{1}{\tau_{\mathrm{m}}}
\begin{pmatrix}
\kappa_{\mathrm{S,S}}(m)\tilde{\alpha}(\nu^{\mathrm{S}}) & \kappa_{\mathrm{S,NS}}(m)\tilde{\alpha}(\nu^{\mathrm{NS}}) \\
\kappa_{\mathrm{NS,S}}(m)\tilde{\alpha}(\nu^{\mathrm{S}}) & \kappa_{\mathrm{NS,NS}}(m)\tilde{\alpha}(\nu^{\mathrm{NS}})
\end{pmatrix}
\tag{11}
$$

from which we obtain the maximum eigenvalue $\rho$, which for stable fixed points must be smaller than 1.

The structure of fixed points for the stimulated sub-population (see discussion in 'Modularity as a bifurcation parameter') can furthermore be intuitively understood by studying the potential landscape of the system. The potential $U$ is thereby defined via the conservative force $F = -\frac{dU}{d\nu^{\mathrm{S}}} = -\nu^{\mathrm{S}} + \nu(\mu, \sigma^2)$ that drives the system toward its fixed points via the equation of motion $\frac{d\nu^{\mathrm{S}}}{dt} = F$ (*Wong and Wang, 2006*; *Litwin-Kumar and Doiron, 2012*; *Schuecker et al., 2017*). Note that $\mu$ and $\sigma^2$ are again functions of $\nu^{\mathrm{S}}$ and $\nu^{\mathrm{NS}}$, where the latter is the self-consistent rate of the non-stimulated sub-populations for given rate $\nu^{\mathrm{S}}$ of the stimulated sub-population, $\nu^{\mathrm{NS}} = \nu^{\mathrm{NS}}(\nu^{\mathrm{S}})$ (details see Appendix B).

## Multiple inputs and correlation-based similarity score

In *Figure 9*, we consider two stimuli $S_1$ and $S_2$ to be active simultaneously for 10 s. Let $SP_1$ and $SP_2$ be the two corresponding SPs in each sub-network. The firing rate of each SP is estimated from spike counts in time bins of 10 ms and smoothed with a Savitzky-Golay filter (length 21 and polynomial order 4). We compute a similarity score based on the correlation between these rates, scaled by the ratio of the input intensities $\lambda_2/\lambda_1$ (with $\lambda_1$ fixed). This scaling is meant to introduce a gradient in

the similarity score based on the firing rate differences, ensuring that high (absolute) scores require comparable activity levels in addition to strong correlations. To ensure that both stimuli are decodable where appropriate, we set the score to 0 when the difference between the rate of $SP_2$ and the non-stimulated SPs was <1 spks/sec ($SP_1$ had significantly higher rates). The curves in *Figure 9c* mark the regime boundaries: coexisting (Co-Ex) where score is >0.1 (red curve); WLC where score is <−0.1 (blue); WTA (gray) and where the score is in the interval (−0.1, 0.1), and either $\lambda_2/\lambda_1 < 0.5$ holds or the score is 0. While the Co-Ex region is a dynamical regime that only occurs in the initial sub-networks (*Figure 9d*), the WTA and WLC regimes persist and can be understood again with the help of a potential $U$, which is in this case a function of the rates of the two SPs (details see Appendix B).

## Numerical simulations and analysis

All numerical simulations were conducted using the Neural Microcircuit Simulation and Analysis Toolkit (NMSAT) v0.2 (*Duarte et al., 2017*), a high-level Python framework for creating, simulating and evaluating complex, spiking neural microcircuits in a modular fashion. It builds on the PyNEST interface for NEST (*Gewaltig and Diesmann, 2007*), which provides the core simulation engine. To ensure the reproduction of all the numerical experiments and figures presented in this study, and abide by the recommendations proposed in *Pauli et al., 2018*, we provide a complete code package that implements project-specific functionality within NMSAT (see Data availability) using NEST 2.18.0 (*Jordan et al., 2019*).

## Competing interests

The authors declare that the research was conducted in the absence of any commercial or financial relationships that could be construed as a potential conflict of interest.

## Acknowledgements

The authors gratefully acknowledge the computing time granted by the JARA-HPC Vergabegremium on the supercomputer JURECA at Forschungszentrum Jülich.

## Additional information

### Funding

| Funder | Grant reference number | Author |
|---|---|---|
| Initiative and Networking Fund of the Helmholtz Association | | Barna Zajzon<br>Abigail Morrison<br>Renato Duarte<br>David Dahmen |
| Helmholtz Portfolio theme Supercomputing and Modeling for the Human Brain | | Barna Zajzon<br>Abigail Morrison<br>Renato Duarte |
| Excellence Initiative of the German federal and state governments | G:(DE-82)EXS-SF-neuroIC002 | Barna Zajzon<br>Abigail Morrison<br>Renato Duarte |
| Helmholtz Association | VH-NG-1028 | David Dahmen |
| European Commission HBP | 945539 | David Dahmen |

The funders had no role in study design, data collection, and interpretation, or the decision to submit the work for publication.

### Author contributions

Barna Zajzon, Conceptualization, Resources, Data curation, Software, Formal analysis, Validation, Investigation, Visualization, Methodology, Writing – original draft, Writing – review and editing; David Dahmen, Conceptualization, Formal analysis, Validation, Investigation, Visualization, Methodology,

Writing – original draft, Writing – review and editing; Abigail Morrison, Conceptualization, Resources, Supervision, Funding acquisition, Investigation, Visualization, Methodology, Writing – original draft, Project administration, Writing – review and editing; Renato Duarte, Conceptualization, Resources, Software, Formal analysis, Supervision, Investigation, Methodology, Writing – original draft, Project administration, Writing – review and editing

## Author ORCIDs
Barna Zajzon  http://orcid.org/0000-0002-3458-103X
David Dahmen  http://orcid.org/0000-0002-7664-916X
Abigail Morrison  http://orcid.org/0000-0001-6933-797X
Renato Duarte  http://orcid.org/0000-0001-6099-667X

## Decision letter and Author response
Decision letter https://doi.org/10.7554/eLife.77009.sa1
Author response https://doi.org/10.7554/eLife.77009.sa2

## Additional files

### Supplementary files
- Supplementary file 1. Tabular description of current-based (baseline) network model.
- Supplementary file 2. Model parameters for the current-based network.
- Supplementary file 3. Parameter values for the conductance-based model.
- Supplementary file 4. Description of the rate model.
- Supplementary file 5. Rate model parameters.
- Transparent reporting form

### Data availability
The current manuscript is a computational study, so no data have been generated for this manuscript. Modelling code can be found at https://doi.org/10.5281/zenodo.6326496 (see also Supplementary Files). Source data and code files are also attached as zip folders to the individual main figures of this manuscript.

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

## Appendix A

### Constraints on feedforward connectivity

This section expands on the limitations arising from the definitions of topographic modularity and map sizes used in this study. By imposing a fixed connection density on the feedforward connection matrices, the projection probabilities between neurons tuned to the same ($p_c$) and different ($p_0$) stimuli are uniquely determined by the modularity $m$ and the parameter $d_0$ and $\delta$, which control the size of stimulus-specific sub-populations (see Materials and methods). For notational simplicity, here we consider the merged excitatory and inhibitory sub-populations tuned to a particular stimulus in a given sub-network SSN$_i$, with a total size $C_i = C_i^E + C_i^I$.

Under the constraints applied in this work, the total density of a feedforward adjacency matrix between SSN$_i$ and SSN$_{i+1}$ can be computed as:

$$\sigma_i = \frac{p_c U_c^i + p_0 U_0^i}{N^2} \tag{12}$$

where $U_0^i$ and $U_c^i$ are the number of *realizable* connections between similarly and differently tuned sub-populations, respectively. Since $U_c^i = N^2 - U_0^i$, we can simplify the notation and focus only on $U_0^i$. We distinguish between the cases of non-overlapping and overlapping stimulus-specific sub-populations:

$$U_0^i = \begin{cases} N^2 - N_C C_i C_{i+1} & \text{if } d_i < \frac{1}{N_C} \\ \frac{N_C}{N_C - 1}(N - C_i)(N - C_{i+1}) & \text{if } d_i \geq \frac{1}{N_C} \end{cases},$$

where each potential synapse is counted only once, regardless of whether the involved neurons belong to any or multiple overlapping sub-populations. This ensures consistency with the definitions of the probabilities $p_c$ and $p_0$. Alternatively, we can express $U_0^i$ as:

$$U_0^i = \frac{N^2 N_{stim}}{N_{stim} - 1}(1 - i\delta - d_0)(1 - (i - 1)\delta - d_0)$$

For the case with no overlap, we can derive an additional constraint on the minimum sub-populations size $C_i$ for the required density $\sigma_i$ to be satisfied, which we define in relation to the total number of sub-populations $N_C$:

$$d_i \geq \sqrt{\frac{\sigma_i}{N_C}} \tag{13}$$

The equality holds in the case of $m = 1$ and all-to-all feedforward connectivity between similarly tuned sub-populations, that is, $p_c = 1$.

## Appendix B

## Mean-field analysis of network dynamics

For an analytical investigation of the role of topographic modularity on the network dynamics, we used mean-field theory (*Fourcaud and Brunel, 2002*; *Helias et al., 2013*; *Schuecker et al., 2015*). Under the assumptions that each neuron receives a large number of small amplitude inputs at every time step, the synaptic time constants $\tau_s$ are small compared to the membrane time constant $\tau_m$, and that the network activity is sufficiently asynchronous and irregular, we can make use of theoretical results obtained from the diffusion approximation of the LIF neuron model to determine the stationary population dynamics. The equations in this section were partially solved using a modified version of the LIF Meanfield Tools library (*Layer et al., 2020*).

### Stationary firing rates and fixed points

In the circumstances described above, the total synaptic input to each neuron can be replaced by a Gaussian white noise process (independent across neurons) with mean $\mu(t)$ and variance $\sigma^2(t)$. In the stationary state, these quantities, along with the firing rates of each afferent, can be well approximated by their constant time average. The stationary firing rate of the LIF neuron in response to such input is:

$$\nu = \left( \tau_{\text{ref}} + \sqrt{\pi} \tau_{\text{eff}} \int_{y_r}^{y_\theta} \exp(u^2) \left[ 1 + \text{erf}\left(u\right) \right] du \right)^{-1} \tag{14}$$

where erf is the error function and the integration limits are defined as $y_r = (V_{\text{reset}} - \mu)/\sigma + \frac{q}{2}\sqrt{\tau_s/\tau_{\text{eff}}}$ and $y_\theta = (\theta - \mu)/\sigma + \frac{q}{2}\sqrt{\tau_s/\tau_{\text{eff}}}$, with $q = \sqrt{2}|\zeta(1/2)|$ and Riemann zeta function $\zeta$ (see *Fourcaud and Brunel, 2002*, Eq. 4.33). As we will see below, the mean μ and variance $\sigma^2$ of the input also depend on the stationary firing rate $\nu$, rendering *Equation 14* an implicit equation that needs to be solved self-consistently using fixed-point iteration.

For simplicity, throughout the mean-field analyses we consider perfectly partitioned networks where each neuron belongs to exactly one topographic map, that is, to one of the $N_C$ stimulus-specific, identically sized sub-populations *SP* (no overlap condition). We denote the firing rate of a neuron in the currently stimulated SP (receiving stimulus input in $\text{SSN}_0$) in sub-network $\text{SSN}_i$ by $\nu_i^S$, and by $\nu_i^{NS}$ that of neurons not associated with the stimulated pathway. Since the firing rates of excitatory and inhibitory neurons are equal (due to identical synaptic time constants and input statistics), we can write the constant mean synaptic input to neurons in the input sub-network as

$$\mu_0^S = \left( \overbrace{K_X J_X \nu_X}^{\text{noise}} + \overbrace{(\frac{1}{N_C} K_E J_E + \frac{1}{N_C} K_I J_I)\nu_0^S}^{\text{rec. stimulated}} + \overbrace{(N_C - 1)(\frac{1}{N_C} K_E J_E + \frac{1}{N_C} K_I J_I)\nu_0^{NS}}^{\text{rec. non-stimulated}} + \overbrace{J_X \nu_{\text{in}}}^{\text{stimulus}} \right) \tau_m$$

$$\mu_0^{NS} = \left( \overbrace{K_X J_X \nu_X}^{\text{noise}} + \overbrace{(\frac{1}{N_C} K_E J_E + \frac{1}{N_C} K_I J_I)\nu_0^S}^{\text{rec. stimulated}} + \overbrace{(N_C - 1)(\frac{1}{N_C} K_E J_E + \frac{1}{N_C} K_I J_I)\nu_0^{NS}}^{\text{rec. non-stimulated}} \right) \tau_m \tag{15}$$

The variances $(\sigma_0^S)^2$ and $(\sigma_0^{NS})^2$ can be obtained by squaring each weight $J$ in the above equation. To derive these equations for the deeper sub-networks $\text{SSN}_{i>0}$, it is helpful to include auxiliary variables $K_S$ and $K_{NS}$, representing the number of feedforward inputs to a neuron in $\text{SSN}_i$ from its own SP in $\text{SSN}_{i-1}$, and from one different SP (there are $N_C - 1$ such sub-populations), respectively. Both $K_S$ and $K_{NS}$ are uniquely defined by the modularity $m$ and projection density $d$, and $K_{NS} = (1 - m)K_S = (1 - m)(1 - \alpha)K_E$ holds as well. The mean synaptic inputs to the neurons in the deeper sub-networks can thus be written as:

$$\mu_i^S = \left( \overbrace{\alpha K_X J_X \nu_X}^{\text{noise}} + \overbrace{(\frac{1}{N_C} K_E J_E + \frac{1}{N_C} K_I J_I) \nu_i^S}^{\text{rec. stimulated}} \right.$$
$$+ \overbrace{(N_C - 1)(\frac{1}{N_C} K_E J_E + \frac{1}{N_C} K_I J_I) \nu_i^{NS}}^{\text{rec. non-stimulated}}$$
$$\left. + \overbrace{K_S J_E \nu_{i-1}^S}^{\text{stimulated FF}} + \overbrace{(N_C - 1) K_{NS} J_E \nu_{i-1}^{NS}}^{\text{non-stimulated FF}} \right) \tau_m \tag{16}$$

$$\mu_i^{NS} = \left( \overbrace{\alpha K_X J_X \nu_X}^{\text{noise}} + \overbrace{(\frac{1}{N_C} K_E J_E + \frac{1}{N_C} K_I J_I) \nu_i^S}^{\text{rec. stimulated}} \right.$$
$$+ \overbrace{(N_C - 1)(\frac{1}{N_C} K_E J_E + \frac{1}{N_C} K_I J_I) \nu_i^{NS}}^{\text{rec. non-stimulated}}$$
$$\left. + K_{NS} J_E \nu_1^S + ((N_C - 2) K_{NS} + K_S) J_E \nu_{i-1}^{NS} \right) \tau_m$$

Again, one can obtain the variances by squaring each weight $J$. The stationary firing rates for the stimulated and non-stimulated sub-populations in all sub-networks are then found by first solving *Equations 14 and 15* for the first sub-network and then (*Equation 16*) *Equations 14 and 16* successively for deeper sub-networks.

For very deep networks, one can ask the question, whether firing rates approach fixed points across sub-networks. If there are multiple fixed points, the initial condition, that is the externally stimulated activity of sub-populations in the first sub-network, decides in which of the fixed points the rates evolve, in a similar spirit as in recurrent networks after a start-up transient. For a fixed point, we have $\nu_{i-1} = \nu_i$. In effect, we can re-group terms in *Equation 16* that have the same rates such that formally we obtain an effective new group of neurons from the excitatory and inhibitory SPs of the current sub-network and the corresponding excitatory SPs of the previous sub-network, as indicated by the square brackets in the following formulas:

$$\mu^S = \alpha \beta \mathcal{J} \nu_X + \mathcal{J} \underbrace{\left[ \frac{1}{N_C} (1 + \gamma g) + (1 - \alpha) \frac{1}{(N_C - 1)(1 - m) + 1} \right]}_{\kappa_{S,S}} \nu^S$$
$$+ \mathcal{J} \underbrace{\left[ \frac{N_C - 1}{N_C} (1 + \gamma g) + (1 - \alpha) \frac{(N_C - 1)(1 - m)}{(N_C - 1)(1 - m) + 1} \right]}_{\kappa_{S,NS}} \nu^{NS} \tag{17}$$

$$\mu^{NS} = \alpha \beta \mathcal{J} \nu_X + \mathcal{J} \underbrace{\left[ \frac{1}{N_C} (1 + \gamma g) + (1 - \alpha) \frac{(1 - m)}{(N_C - 1)(1 - m) + 1} \right]}_{\kappa_{NS,S}} \nu^S$$
$$+ \mathcal{J} \underbrace{\left[ \frac{N_C - 1}{N_C} (1 + \gamma g) + (1 - \alpha) \frac{1 + (N_C - 2)(1 - m)}{(N_C - 1)(1 - m) + 1} \right]}_{\kappa_{NS,NS}} \nu^{NS} \tag{18}$$

with $\beta = K_X / K_E$, $\gamma = K_I / K_E$, and $\mathcal{J} = \tau K_E J$.
For the parameters $g$ and $\gamma$ chosen here, $\kappa_{S,NS}$, $\kappa_{NS,S}$, and $\kappa_{NS,NS}$ in *Equations 17 and 18* are always negative for any modularity $m$ due to the large recurrent inhibition. Therefore, for the non-stimulated group, $\kappa < 0$ in *Equation 8* (see main text), such that one always finds a single fixed point, which, as desired, is at a low rate. Interestingly, the excitatory feedforward connections can switch the sign of $\kappa_{S,S}$ from negative to positive for large values of $m$, thereby rendering the active group effectively excitatory, leading to a saddle-node bifurcation and the emergence of a stable high-activity fixed point (see *Figure 7b* in the main text).

The structure of fixed points can also be understood by studying the potential landscape of the system: *Equation 14* can be regarded as the fixed-point solution of the following evolution equations for the stimulated and non-stimulated sub-populations (*Wong and Wang, 2006*; *Schuecker et al., 2017*)

$$\tau_S \frac{d\nu^S}{dt} = -\nu^S + \Phi_S(\nu^S, \nu^{NS}),\tag{19}$$

$$\tau_{NS} \frac{d\nu^{NS}}{dt} = -\nu^{NS} + \Phi_{NS}(\nu^S, \nu^{NS}),\tag{20}$$

where $\Phi_S$ and $\Phi_{NS}$ are defined via the right-hand side of *Equation 14* with $\mu^S$ and $\mu^{NS}$ inserted as defined in *Equations 17 and 18* (and likewise for $\sigma^S$ and $\sigma^{NS}$). Due to the asymmetry in connections between stimulated and non-stimulated sub-populations, the right-hand side of *Equations 19 and 20* cannot be interpreted as a conservative force. Following the idea of effective response functions (*Mascaro and Amit, 1999*), a potential $U(\nu^S)$ for the stimulated sub-population alone can, however, be defined by inserting the solution $\nu^{NS} = f(\nu^S)$ of *Equation 20* into *Equation 19*

$$\tau_S \frac{d\nu^S}{dt} = -\nu^S + \Phi_S(\nu^S, f(\nu^S))\tag{21}$$

and interpreting the right-hand side as a conservative force $F = -\frac{dU}{d\nu^S}$ (*Litwin-Kumar and Doiron, 2012*). The potential then follows from integration as

$$U(\nu^S) - U(0) = \frac{1}{2}(\nu^S)^2 - \int_0^{\nu^S} \Phi_S(\nu, f(\nu))d\nu,\tag{22}$$

where $U(0)$ is an inconsequential constant. We solved the latter integral numerically using the scipy.integrate.trapz function of SciPy (*Virtanen et al., 2020*). The minima and maxima of the resulting potential correspond to locally stable and unstable fixed points, respectively. Note that while this single-population potential is useful to study the structure of fixed points, the full dynamics of all populations and global stability cannot be straight-forwardly infered from this reduced picture (*Mascaro and Amit, 1999*; *Rost et al., 2018*), here for two reasons: (1) For spiking networks, *Equation 19* and *Equation 20* do not describe the real dynamics of the mean activity. Their right-hand side only defines the stationary state solution. (2) The global stability of fixed points also depends on the time constants of all sub-populations' mean activities (here $\tau_S$ and $\tau_{NS}$), but the temporal dynamics of the non-stimulated sub-populations is neglected here.

## Mean-field analysis for two input streams

In the case of two simultaneously active stimuli (see Section 'Input integration and multi-stability'), if the stimulated group 1 is in the high-activity state with rate $\nu^{S1}$, the second stimulated group 2 will receive an additional non-vanishing input of the form

$$\left[\frac{1}{N_C}(1 + \gamma g) + (1 - \alpha)\frac{(1-m)}{(N_C-1)(1-m)+1}\right]\nu^{S1} < 0,\tag{23}$$

which is negative for all values of $m$ and can therefore lead to the silencing of group 2. If the stimuli are similarly strong, network fluctuations can dynamically switch the roles of the stimulated groups 1 and 2.

The dynamics and fixed-point structure in deep sub-networks can be studied using a two-dimensional potential landscape that is defined via the following evolution equation

$$\frac{d\nu^{S1}}{dt} = -\nu^{S1} + \Phi_{S1}(\nu^{S1}, \nu^{S2}, f(\nu^{S1}, \nu^{S2})),\tag{24}$$

$$\frac{d\nu^{S2}}{dt} = -\nu^{S2} + \Phi_{S2}(\nu^{S1}, \nu^{S2}, f(\nu^{S1}, \nu^{S2})),\tag{25}$$

where $f(\nu^{S1}, \nu^{S2}) = \nu^{NS}$ is the fixed point of the non-stimulated sub-populations for given rates $\nu^{S1}, \nu^{S2}$ of the two stimulated sub-populations, respectively. The functions $\Phi_{S1}$ and $\Phi_{S2}$ are again defined via the right-hand side of *Equation 14* with inserted $\mu^{S1}$, $\mu^{S2}$ and $\mu^{NS}$ that are defined as follows (derivation analogous to the single-input case):

$$\mu^{\mathrm{S1}} = \alpha \mathcal{J} \nu_{\mathrm{X}} + \mathcal{J} \underbrace{\left[ \frac{1}{N_{\mathrm{C}}} \left(1 + \gamma g\right) + (1 - \alpha) \frac{1}{(N_{\mathrm{C}} - 1)(1 - m) + 1} \right]}_{\kappa_{\mathrm{S1,S1}}} \nu^{\mathrm{S1}}$$

$$+ \mathcal{J} \underbrace{\left[ \frac{1}{N_{\mathrm{C}}} \left(1 + \gamma g\right) + (1 - \alpha) \frac{1 - m}{(N_{\mathrm{C}} - 1)(1 - m) + 1} \right]}_{\kappa_{\mathrm{S1,S2}}} \nu^{\mathrm{S2}} \qquad (26)$$

$$+ \mathcal{J} \underbrace{\left[ \frac{N_{\mathrm{C}} - 2}{N_{\mathrm{C}}} \left(1 + \gamma g\right) + (1 - \alpha) \frac{(N_{\mathrm{C}} - 2)(1 - m)}{(N_{\mathrm{C}} - 1)(1 - m) + 1} \right]}_{\kappa_{\mathrm{S1,NS}}} \nu^{\mathrm{NS}}$$

$$\mu^{\mathrm{S2}} = \alpha \mathcal{J} \nu_{\mathrm{X}} + \mathcal{J} \underbrace{\left[ \frac{1}{N_{\mathrm{C}}} \left(1 + \gamma g\right) + (1 - \alpha) \frac{1 - m}{(N_{\mathrm{C}} - 1)(1 - m) + 1} \right]}_{\kappa_{\mathrm{S2,S1}}} \nu^{\mathrm{S1}}$$

$$+ \mathcal{J} \underbrace{\left[ \frac{1}{N_{\mathrm{C}}} \left(1 + \gamma g\right) + (1 - \alpha) \frac{1}{(N_{\mathrm{C}} - 1)(1 - m) + 1} \right]}_{\kappa_{\mathrm{S2,S2}}} \nu^{\mathrm{S2}} \qquad (27)$$

$$+ \mathcal{J} \underbrace{\left[ \frac{N_{\mathrm{C}} - 2}{N_{\mathrm{C}}} \left(1 + \gamma g\right) + (1 - \alpha) \frac{(N_{\mathrm{C}} - 2)(1 - m)}{(N_{\mathrm{C}} - 1)(1 - m) + 1} \right]}_{\kappa_{\mathrm{S1,NS}}} \nu^{\mathrm{NS}}$$

$$\mu^{\mathrm{NS}} = \alpha \mathcal{J} \nu_{\mathrm{X}} + \mathcal{J} \underbrace{\left[ \frac{1}{N_{\mathrm{C}}} \left(1 + \gamma g\right) + (1 - \alpha) \frac{(1 - m)}{(N_{\mathrm{C}} - 1)(1 - m) + 1} \right]}_{\kappa_{\mathrm{NS,S1}}} \nu^{\mathrm{S1}} \qquad (28)$$

$$+ \mathcal{J} \underbrace{\left[ \frac{1}{N_{\mathrm{C}}} \left(1 + \gamma g\right) + (1 - \alpha) \frac{(1 - m)}{(N_{\mathrm{C}} - 1)(1 - m) + 1} \right]}_{\kappa_{\mathrm{NS,S2}}} \nu^{\mathrm{S2}}$$

$$+ \mathcal{J} \underbrace{\left[ \frac{N_{\mathrm{C}} - 2}{N_{\mathrm{C}}} \left(1 + \gamma g\right) + (1 - \alpha) \frac{1 + (N_{\mathrm{C}} - 3)(1 - m)}{(N_{\mathrm{C}} - 1)(1 - m) + 1} \right]}_{\kappa_{\mathrm{NS,NS}}} \nu^{\mathrm{NS}} \qquad (29)$$

Due to the symmetry between the two stimulated sub-populations, the right-hand side of *Equations 24 and 25* can be viewed as a conservative force $\boldsymbol{F}$ of the potential $U(\nu^{\mathrm{S1}}, \nu^{\mathrm{S2}}) = - \int_{\mathcal{C}} \boldsymbol{F}\, ds$, where we parameterized the line integral along the path $\nu : [0, 1] \to \mathcal{C}, t \mapsto t \cdot (\nu^{\mathrm{S1}}, \nu^{\mathrm{S2}})$, which yields

$$U(\nu^{\mathrm{S1}}, \nu^{\mathrm{S2}}) = \frac{1}{2}(\nu^{\mathrm{S1}})^2 + \frac{1}{2}(\nu^{\mathrm{S2}})^2 - \int_0^{\nu^{\mathrm{S1}}} \Phi_{\mathrm{S1}} \left( \nu, \nu \frac{\nu^{\mathrm{S2}}}{\nu^{\mathrm{S1}}}, f(\nu, \nu \frac{\nu^{\mathrm{S2}}}{\nu^{\mathrm{S1}}}) \right) - \int_0^{\nu^{\mathrm{S2}}} \Phi_{\mathrm{S2}} \left( \nu \frac{\nu^{\mathrm{S1}}}{\nu^{\mathrm{S2}}}, \nu, f(\nu \frac{\nu^{\mathrm{S1}}}{\nu^{\mathrm{S2}}}, \nu) \right) .$$

$$(30)$$

The numerical evaluation of this two-dimensional potential is shown in *Figure 9—figure supplement 2*, whereas sketches in *Figure 9e* show a one-dimensional section (gray lines in *Figure 9—figure supplement 2*) that goes anti-diagonal through the two minima corresponding to one population being in the high-activity state and the other one being in the low-activity state.

## Critical modularity for piecewise linear activation function

To obtain a closed-form analytic solution for the critical modularity, in the following we consider a neuron model with piecewise linear activation function

$$\nu(\mu) = \nu_{\max} \frac{\mu - \mu_{\min}}{\mu_{\max} - \mu_{\min}} \qquad (31)$$

for $\mu \in [\mu_{\min}, \mu_{\max}]$, $\nu(\mu) = 0$ for $\mu < \mu_{\min}$ and $\nu(\mu) = \nu_{\max}$ for $\mu > \mu_{\max}$ (*Figure 8a*). Successful denoising requires the non-stimulated sub-populations to be silent, $\nu^{NS} = 0$, and the stimulated sub-populations to be active, $\nu^{S} > 0$. We first study solutions where $0 < \nu^{S} < \nu_{\max}$ and afterwards the case where $\nu^{S} = \nu_{\max}$. Inserting *Equation 31* into *Equations 9 and 10*, we obtain

$$\mu^{\mathrm{S}} = \alpha \mathcal{J} \nu_{\mathrm{X}} + \kappa_{\mathrm{S,S}}(m)\, \nu_{\max} \frac{\mu_S - \mu_{\min}}{\mu_{\max} - \mu_{\min}} ,$$

$$\mu^{\mathrm{NS}} = \alpha \mathcal{J} \nu_{\mathrm{X}} + \kappa_{\mathrm{NS,S}}(m)\, \nu_{\max} \frac{\mu_S - \mu_{\min}}{\mu_{\max} - \mu_{\min}} .$$

The first equation can be solved for $\mu^S$

$$\frac{\mu^S}{\mu_{min}} = 1 + \frac{\alpha \mathcal{J} \nu_X - \mu_{min}}{\mu_{min} - \kappa_{S,S}(m) \nu_{max} \frac{\mu_{min}}{\mu_{max} - \mu_{min}}}, \tag{32}$$

which holds for

$$\mu_{min} \leq \mu^S \leq \mu_{max}, \tag{33}$$

$$\mu^{NS} \leq \mu_{min}. \tag{34}$$

Requirement (*Equation 33*) is equivalent to an inequality for $m$

$$0 \leq \frac{\alpha \mathcal{J} \nu_X - \mu_{min}}{\mu_{max} - \frac{\mathcal{J}}{N_C}(1+\gamma g)\nu_{max} - \frac{(1-\alpha)\mathcal{J}\nu_{max}}{(N_C-1)(1-m)+1} - \mu_{min}} \leq 1$$

that, depending on the dynamic range of the neuron, the strength of the external background input and the recurrence, yields

$$m = \frac{N_C}{N_C - 1} - \frac{1}{N_C - 1} \frac{(1-\alpha)\mathcal{J}\nu_{max}}{\mu_{max} - \alpha \mathcal{J} \nu_X - \frac{\mathcal{J}}{N_C}(1+\gamma g)\nu_{max}} \tag{35}$$

as an upper or lower bound for the modularity (*Figure 8*). Requirement (*Equation 34*) with the solution (*Equation 32*) for $\mu^S$ inserted yields a further lower bound

$$m \geq \frac{(\mu_{max} - \mu_{min})N_C}{(1-\alpha)\mathcal{J}\nu_{max} + (\mu_{max} - \mu_{min})(N_C - 1)} \tag{36}$$

for the modularity that is required for denoising. This criterion is independent of the external background input and the recurrence of the SSN.

Now we turn to the saturated scenario $\nu^S = \nu_{max}$ and $\nu^{NS} = 0$ and obtain

$$\mu^S = \alpha \mathcal{J} \nu_X + \kappa_{S,S}(m) \nu_{max},$$
$$\mu^{NS} = \alpha \mathcal{J} \nu_X + \kappa_{NS,S}(m) \nu_{max},$$

with the criteria

$$\mu^S \geq \mu_{max}, \tag{37}$$

$$\mu^{NS} \leq \mu_{min}. \tag{38}$$

The first criterion (*Equation 37*) yields the same critical value (*Equation 35*) that for $\mu_{max} - \alpha \mathcal{J} \nu_X - \frac{\mathcal{J}}{N_C}(1+\gamma g)\nu_{max} \geq 0$ is a lower bound and otherwise an upper bound. The second criterion (*Equation 38*) yields an additional lower bound for $\mathcal{J}(1-\alpha)\nu_{max} - (N_C - 1)\left(\mu_{min} - \alpha \mathcal{J} \nu_X - \frac{\mathcal{J}}{N_C}(1+\gamma g)\nu_{max}\right) \geq 0$ (*Figure 8*):

$$m \geq 1 - \frac{\left(\mu_{min} - \alpha \mathcal{J} \nu_X - \frac{\mathcal{J}}{N_C}(1+\gamma g)\nu_{max}\right)}{\mathcal{J}(1-\alpha)\nu_{max} - (N_C-1)\left(\mu_{min} - \alpha \mathcal{J} \nu_X - \frac{\mathcal{J}}{N_C}(1+\gamma g)\nu_{max}\right)}. \tag{39}$$

The above criteria yield necessary conditions for the existence of a fixed point with $\nu^S > 0$ and $\nu^{NS} = 0$. Next we study the stability of such solutions. This works analogous to the stability in the spiking models discussed in Section 'Effective connectivity and stability analysis' by studying the spectrum of the effective connectivity matrix. For the model *Equation 31*, the effective connectivity is given by

$$w_{ij} = \frac{\partial \nu_i}{\partial \nu_j} = \nu'(\mu_i)\frac{\partial \mu_i}{\partial \nu_j} = \nu'(\mu_i)\mathcal{J}_{ij} \tag{40}$$

with $\nu'(\mu) = \frac{d\nu}{d\mu}(\mu)$ and $\mathcal{J}_{ij} = \tau_x J_{ij}$. On the level of stimulated and non-stimulated sub-populations across layers, the effective connectivity becomes

$$W = \begin{pmatrix} \kappa_{S,S}(m)\nu'(\mu^S) & \kappa_{S,NS}(m)\nu'(\mu^{NS}) \\ \kappa_{NS,S}(m)\nu'(\mu^S) & \kappa_{NS,NS}(m)\nu'(\mu^{NS}) \end{pmatrix} \tag{41}$$

with eigenvalues

$$\begin{aligned} \lambda_\pm \quad = \quad & \frac{\kappa_{S,S}(m)\nu'(\mu^S) + \kappa_{NS,NS}(m)\nu'(\mu^{NS})}{2} \\ & \pm \sqrt{\left(\frac{\kappa_{S,S}(m)\nu'(\mu^S) + \kappa_{NS,NS}(m)\nu'(\mu^{NS})}{2}\right)^2 - \left(\kappa_{S,S}(m)\nu'(\mu^S)\kappa_{NS,NS}(m)\nu'(\mu^{NS}) - \kappa_{S,NS}(m)\nu'(\mu^{NS})\kappa_{NS,S}(m)\nu'(\mu^S)\right)}. \end{aligned} \tag{42}$$

The saturated fixed point $\nu^S = \nu_{max}$ and $\nu^{NS} = 0$ has $\nu'(\mu^S) = \nu'(\mu^{NS}) = 0$, leading to $\lambda_\pm = 0$. This fixed point is always stable. The non-saturated fixed point also has $\nu'(\mu^{NS}) = 0$. Consequently, *Equation 42* simplifies to $\lambda_- = 0$ and

$$\lambda_+ = \frac{\nu_{max}}{\mu_{max} - \mu_{min}} \kappa_{S,S}(m). \tag{43}$$

For $\lambda > 1$ fluctuations in the stimulated sub-population are being amplified. These fluctuations also drive fluctuations of the non-stimulated sub-population via the recurrent coupling. The fixed point thus becomes unstable and the necessary distinction between the stimulated and non-stimulated sub-populations vanishes. For inhibition-dominated recurrence, $\kappa_{S,S}(m)$ is small enough to obtain stable fixed points at non-saturated rates (*Figure 8c*). In the case of no recurrence or excitation-dominated recurrence, $\kappa_{S,S}(m)$ is much larger, typically driving $\lambda_+$ across the line of instability and preventing non-saturated fixed points to be stable. In such networks, only the saturated fixed point at $\nu^S = \nu_{max}$ is stable and reachable (*Figure 8d and e*).

