## [Editor Report]

This manuscript puts forward a new idea that topography in neural networks helps to remove noise from inputs. The authors show that there is a critical level of topography that is needed for network to denoise inputs.

---

## [Decision Letter]

**Decision letter after peer review:**

Thank you for submitting your article "Signal denoising through topographic modularity of neural circuits" for consideration by *eLife*. Your article has been reviewed by 2 peer reviewers, one of whom is a member of our Board of Reviewing Editors, and the evaluation has been overseen by Joshua Gold as the Senior Editor. The reviewers have opted to remain anonymous.

Essential revisions:

To increase the impact of this work, it is necessary to

1) Clarify the properties of the critical point (see comments from Reviewer 1).

2) Consider how denoising could work for dynamic inputs (comments raised by both Reviewers).

*Reviewer #1 (Recommendations for the authors):*

As I have stated in the main review section, my main issue with this work is that I fail to see the impact of the results. The authors provide a detailed analysis of a model for cortical connectivity with topographical connections. On the one side, they do not compare their analysis to neuronal recording or imaging, thus not providing evidence that their analysis of the dynamics is correct and justifies the model. On the other hand, the model does not offer deep theoretical insights and focuses on simplistic computational tasks, denoising, which could be achieved in different ways.

In the following, I give some recommendations to the authors on making the work more meaningful and robust, in my opinion. First, I will address the bigger question of what could be done to increase the impact of the work. Then, I will address some technical issues and lack of coherence.

– I would have liked to see a theoretical derivation of the critical modularity level. For example, the authors can attempt to derive bifurcation curves in Figure 7 and show how they depend on different parameters in the system. The authors show that the single neuron dynamics do not affect the result but other connectivity parameters (e.g., the mean and variance of the different weight matrices). If the authors think that m=0.83 is a universal critical value, they should argue for it.

– One possible way to better understand the dynamics in the network and the role of the recurrent connectivity is to extend the mean-field analysis to the fluctuations around the fixed points of each population. For example, one concern is that while the mean activity would be low in inactive channels, noise fluctuations could still propagate.

– Can the results be compared to a more simplified feedforward network with topographic connectivity? Is the recurrent connectivity needed to explain and interpret the result? Is the inhibitory network required, or can similar effects be achieved with a subcritical excitatory recurrent population in each layer?

*Reviewer #2 (Recommendations for the authors):*

The manuscript presents an interesting and novel idea. My main suggestions for improvement pertain to the clarity of the presentation. In many cases the results are presented out-of-order and it is difficult to understand the authors point until reading the next paragraph.

For example, the critical value is mentioned in the first paragraph on page 4, but at that point it is not explained that there is a transition. On page 5, the discussion of Figure 2 returns back to Figure 1. It might be better to re-order the panels within figures to ensure continuous sequential description.

An important typo in Figure 2 legend "For" should be "Four".

---

## [Author Response]

Reviewer #1 (Recommendations for the authors):As I have stated in the main review section, my main issue with this work is that I fail to see the impact of the results. The authors provide a detailed analysis of a model for cortical connectivity with topographical connections. On the one side, they do not compare their analysis to neuronal recording or imaging, thus not providing evidence that their analysis of the dynamics is correct and justifies the model. On the other hand, the model does not offer deep theoretical insights and focuses on simplistic computational tasks, denoising, which could be achieved in different ways.In the following, I give some recommendations to the authors on making the work more meaningful and robust, in my opinion. First, I will address the bigger question of what could be done to increase the impact of the work. Then, I will address some technical issues and lack of coherence.

We thank the reviewer for the constructive feedback and valuable comments. We believe that they helped us to improve the quality and impact of our study significantly.

– I would have liked to see a theoretical derivation of the critical modularity level. For example, the authors can attempt to derive bifurcation curves in Figure 7 and show how they depend on different parameters in the system. The authors show that the single neuron dynamics do not affect the result but other connectivity parameters (e.g., the mean and variance of the different weight matrices). If the authors think that m=0.83 is a universal critical value, they should argue for it.

We apologize for the misunderstanding that the single-neuron dynamics does not affect the critical value of the modularity. The conductance-based and current-based spiking model are very similar in their dynamic behavior, leading to a similar value for the transition/switching modularity (*m* = 0*.*83 vs *m* = 0*.*85). We adapted the text to clarify this difference (see lines 208-211 and 274-276), and provided a more in-depth analysis of the influence of single-neuron properties, recurrent connectivity and inhibition on the critical modularity (see comment above, section ”Critical modularity for denoising” and Appendix B). Furthermore, we apologize for not having made clear in the previous version that the bifurcation curves in Figure 7 were indeed theoretically derived. The mean-field self-consistent equations, however, had to be solved numerically due to the complicated nonlinear activation function of spiking neurons. In the revised manuscript, we derived fully analytic closed-form expressions for the critical modularity for a qualitatively similar but more tractable piecewise linear activation function. These expressions are helpful to understand the role of recurrence, external input and single-neuron properties for denoising, see new section ”Critical modularity for denoising”.

– One possible way to better understand the dynamics in the network and the role of the recurrent connectivity is to extend the mean-field analysis to the fluctuations around the fixed points of each population. For example, one concern is that while the mean activity would be low in inactive channels, noise fluctuations could still propagate.

We thank the reviewer for this great suggestion and added a fluctuation analysis for the fully analytically tractable piecewise linear neuron model (Figure 8). The analysis shows that recurrent inhibition stabilizes both the high activity fixed point of the stimulated sub-population and the low activity fixed point of the non-stimulated sub-populations, yielding robust denoising behavior. This feature is lost for networks without recurrent connections or excitation-dominated networks, see new section ”Critical modularity for denoising”.

– Can the results be compared to a more simplified feedforward network with topographic connectivity? Is the recurrent connectivity needed to explain and interpret the result? Is the inhibitory network required, or can similar effects be achieved with a subcritical excitatory recurrent population in each layer?

In the revised manuscript, we address these questions in the new Figure 8 and section ”Critical modularity for denoising”. The results show that for biologically plausible settings (excitatory background and feedforward input) recurrent inhibition is crucial for denoising. We also study the case of no recurrence (purely feedforward network) and excitation-dominated recurrent connectivity and theoretically predict that no denoising can be achieved assuming biological constraints. We further validate these predictions with numerical simulations of corresponding spiking networks.

Reviewer #2 (Recommendations for the authors):The manuscript presents an interesting and novel idea. My main suggestions for improvement pertain to the clarity of the presentation. In many cases the results are presented out-of-order and it is difficult to understand the authors point until reading the next paragraph.

We thank the reviewer for the helpful suggestions and apologize for the lack of clarity in our initial submission. We have taken the reviewer’s suggestions into consideration and hope the manuscript is now clearer and more understandable.

For example, the critical value is mentioned in the first paragraph on page 4, but at that point it is not explained that there is a transition.

Please note that in the revised submission, we differentiate more explicitly between the transition point *m*_switch_ (dependent on the input) and the critical modularity threshold *m*_crit_ (independent of input). As the reviewer points out, the critical modularity which, in fact, referred to the switching point *m*_switch_, was first mentioned in the initial paragraph of page 4. In this same paragraph of the revised text, we make it explicit that this represents a qualitative transition in the behavior of the system: ”However, as *m* approaches a switching value *m*_switch_ ≈ 0*.*83, there is a qualitative transition in the system’s behavior, leading to a consistently higher reconstruction accuracy across the sub-networks

The role of *m*_switch_ as a transition point is further clarified in the beginning of the next paragraph:

“Beyond this transition point, reconstruction accuracy improves with depth”

We believe this makes it clear that the switching value *m*_switch_ represents a qualitative transition point, but if the reviewer disagrees, we would welcome suggestions on how to improve the clarity and readability of this section. For a detailed discussion on the related critical modularity threshold *m*_crit_, see new section ”Critical modularity for denoising”.

On page 5, the discussion of Figure 2 returns back to Figure 1. It might be better to re-order the panels within figures to ensure continuous sequential description.

We thank the reviewer for pointing this out. It is indeed a good practice to ensure sequential description and make sure the different sections refer to their corresponding figures / panels in a sequential order. After a careful revision, we have concluded that the reference to Figure 1 does not break the sequentiality of the results’ presentation. Figure 1 presents the model setup and emphasizes task performance whereas Figure 2 focuses on network dynamics. We concluded this order of presentation is clearest and the most aligned with the main text. When presenting the results of Figure 2, it is important to relate the system’s dynamics to the computational performance instead of treating them in isolation. For that reason, the reference to ”the zero-gain convergence point in Figure 1d, g” is pertinent. We do not believe this presentation of the results breaks the flow or constitutes a discontinuity, but if the reviewer disagrees, we welcome suggestions on how this could be improved.

An important typo in Figure 2 legend "For" should be "Four".

We apologize if we misunderstood the reviewer, but we could not find any typo or discrepancy. We believe the reviewer refers to the following sentence:

”For modularity values facilitating an asynchronous irregular regime across the network, ”

The use of ”For” here is correct, and refers to all the modularity values depicted in panel (c) which facilitate an asynchronous irregular activity. We verified all other occurrences of ”for” and found no errors (none of the panels contain or illustrate exactly four values / elements). If we overlooked something, we would kindly ask the reviewer to direct our attention to what exactly is considered a typo in this legend.

References

Massimo Mascaro and Daniel J Amit. Effective neural response function for collective population states. *Network: Computation in Neural Systems*, 10(4):351–373, 1999.

Thomas Rost, Moritz Deger, and Martin P Nawrot. Winnerless competition in clustered balanced networks: inhibitory assemblies do the trick. *Biological cybernetics*, 112(1):81–98, 2018.